# ECAttention: Adaptive and Principled Feature Expansion-Compression with Linear Efficiency

## Abstract

The maximal coding rate reduction ($MCR^2$) objective has been proposed to learn low-dimensional subspace representations by minimizing the compression term for intra-group compactness and maximizing the expansion term for inter-group separation. Several studies have leveraged $MCR^2$ to design principled, interpretable deep models by following or approximating its gradient to derive layer structures. However, these approaches remain limited in achieving fully principled and effective compression and lack self-adaptive control over the strength of expansion and compression across layers. In this work, we introduce **ECAttention**, a novel attention mechanism that incorporates principled expansion and compression modules inspired by the **geometric insight** of $MCR^2$. Geometrically, gradient-based updates of $MCR^2$ move features along directions shaped by the underlying data structure. Our method efficiently captures this structure using randomization combined with Cholesky decomposition to guide feature updates with **nearly linear complexity**. By introducing two trainable weights per layer, ECAttention self-adaptively regulates the strengths of compression and expansion. The resulting ECA transformer not only matches or surpasses prior methods, but also exhibits greater interpretability, with different heads focusing on distinct image regions and capturing **fine-grained structures** under simple supervised training.

## 1 Introduction

Recent studies reveal that deep learning models exhibit feature expansion in the early layers, promoting inter-class feature separation and facilitating intra-class feature compression in deeper layers (Xu et al., 2025; Alain & Bengio, 2016; Masarczyk et al., 2023; Wang et al., 2025). Unlike traditional deep learning models, ReduNet (Chan et al., 2022) provides a principled framework that explicitly implements modules for both feature expansion and compression in each layer to learn low-dimensional, class-specific subspace representations. This is achieved by maximizing the *maximal coding rate reduction* ($MCR^2$) objective, which combines compression and expansion terms, with each layer's operators derived in a "white-box" manner from its gradient ascent updates. Specifically, the compression operator minimizes the intra-class coding rate to shrink the volume of each class subspace, while the expansion operator maximizes the overall coding rate to enlarge the entire feature space and drive class subspaces toward orthogonality, potentially maximizing inter-class separation. Two variants of ReduNet have been introduced, both designed to maintain its principled architecture while enhancing scalability. Coding-RATE transformer (CRATE) approximates the gradient descent step on a modified compression term of $MCR^2$, yielding a multi-head subspace self-attention operator (MSSA) aimed at minimizing this objective (Yu et al., 2023). Token statistics transformer (ToST) further reduces computational complexity by adopting an upper-bound of the modified compression term that depends only on the diagonal elements of the feature correlation matrix (Wu et al., 2025). By applying a gradient descent step on this variant, it derives a linear-time attention operator, termed token statistics self-attention (TSSA).

However, these approaches suffer from two main limitations. First, both CRATE and ToST fail to achieve effective and fully principled compression: MSSA contradicts its theoretical design, as it increases rather than decreases the compression term (Hu et al., 2024). TSSA fails to effectively

capture fine-grained intrinsic structures of data, as it solely relies on the diagonal of the feature correlation matrix, neglecting correlations among dimensions. These limitations ultimately undermine the interpretability of the constructed network model. Second, ReduNet and its variants lack a self-adaptive strategy to balance expansion and compression across layers. Yu et al. (2024) show that prioritizing expansion in the early layers of ReduNet can produce more compact feature representations and significantly improve the model performance.

In this work, we revisit MCR$^2$ from a **geometric perspective** and demonstrate that effective compression can be achieved by narrowing the dimensions that have irrelevant information to reduce the volume of the subspace, rather than simply modifying values within a fixed-dimensional setting. Building on this observation, we propose **ECAttention**, a novel attention operator that expands and compresses features by adjusting the dimensionality of the feature matrix's column space, rather than relying on the gradient. Compared with existing approaches, ECAttention offers three key advantages: (1) **Effective and principled compression:** we show that the resulting ECA transformer achieves better interpretability. This is reflected not only in the principled design of the modules, but also in the fact that different heads across layers learn to attend to distinct, fine-grained regions of images solely from simple classification tasks. We leverage a randomization method to efficiently extract the most informative dimensions of the representation space while also reducing dimensionality. We then apply Cholesky decomposition to this reduced space to obtain its basis, which becomes the foundation for constructing our expansion and compression modules. (2) **Self-adaptive expansion and compression:** we equip the expansion and compression modules in each layer with individual trainable scaling factors, allowing their weights to be automatically balanced through backpropagation. We show that ECAttention exhibits self-adaptive capabilities: it learns to prioritize feature expansion in early layers when the compressed space is small, as evidenced by the weight variation patterns. In contrast, the expansion module tends to compress the entire feature space when the compressed space is large. (3) **Linear complexity**: ECAttention achieves linear complexity in both time and memory, scaling as $\mathcal{O}(n)$ where $n$ is the number of tokens, with a minor fixed overhead due to the Cholesky decomposition.

Our experiments demonstrate that ECA achieves comparable or superior performance to ToST and CRATE, while maintaining low computational overhead, memory usage, and parameter count. In particular, despite its linear complexity and only 7.2M parameters, ECA-S achieves 79.0% top-1 accuracy on ImageNet ReaL, surpassing CRATE-L (77.6M parameters, quadratic complexity) which attains 77.4%. Moreover, ECA delivers performance comparable to ToST while producing more interpretable and fine-grained attention maps.

## 2 BACKGROUND AND RELATED WORK

**Notation.** For a vector $\boldsymbol{v} \in \mathbb{R}^n$, let $\mathrm{Diag}(\boldsymbol{v}) \in \mathbb{R}^{n \times n}$ be the diagonal matrix with $\mathrm{Diag}(\boldsymbol{v})_{ii} = \boldsymbol{v}_i$. Let $\boldsymbol{I}$ be the identity matrix. For a positive integer $n$, let $[n] = \{1, 2, \ldots, n\}$. We denote by $\mathbf{1}$ the vector of all ones. We denote by $\mathrm{PSD}(n) \subseteq \mathbb{R}^{n \times n}$ the set of $n \times n$ positive semi-definite (PSD) matrices. For $\boldsymbol{M} \in \mathrm{PSD}(n)$ and $i \in [n]$, we denote by $\lambda_i(\boldsymbol{M})$ the $i$-th largest eigenvalue of $\boldsymbol{M}$.

Raw data such as images are typically first tokenized into vectors $\boldsymbol{X} = [\boldsymbol{x}_1, \ldots, \boldsymbol{x}_n] \in \mathbb{R}^{D \times n}$ for training, where each vector $\boldsymbol{x}_i \in \mathbb{R}^D, i \in [n]$ represents a local part of the data (e.g., a patch of an image) (Dosovitskiy et al., 2021). These tokens often belong to different semantic categories, and conventional deep models mainly aim to learn task-specific representations for prediction. In contrast, ReduNet introduces the MCR$^2$ objective, which instead focuses on learning task-agnostic subspace representations $\boldsymbol{Z} = [\boldsymbol{z}_1, \ldots, \boldsymbol{z}_n] \in \mathbb{R}^{d \times n}$ from tokens $\boldsymbol{X}$ (Chan et al., 2022). Specifically, tokens (or patches) of an image are assumed to belong to $K$ groups, and MCR$^2$ aims to learn token features that are compact within each group while being well separated across $K$ groups. Let $\boldsymbol{\Pi} = [\boldsymbol{\pi}_1, \ldots, \boldsymbol{\pi}_K] \in \mathbb{R}^{n \times K}$ be a membership matrix, where the $k$-th column $\boldsymbol{\pi}_k \in \mathbb{R}^n$ contains the soft assignment weights of all $n$ tokens to the $k$-th group. For every $i \in [n]$, we have $\sum_{j \in [K]} \boldsymbol{\Pi}_{ij} = 1$. Let $\epsilon > 0$ and $n_k = \langle \boldsymbol{\pi}_k, \mathbf{1} \rangle$ for each $k \in [K]$. The MCR$^2$ is defined as follows:

$$\Delta R(\boldsymbol{Z}, \boldsymbol{\Pi}) = \underbrace{\frac{1}{2} \log \det(\boldsymbol{I} + \frac{d}{n\epsilon^2} \boldsymbol{Z}\boldsymbol{Z}^T)}_{R(\boldsymbol{Z})} - \underbrace{\frac{1}{2} \sum_{k=1}^{K} \frac{n_k}{n} \log \det(\boldsymbol{I} + \frac{d}{n_k\epsilon^2} \boldsymbol{Z}\mathrm{Diag}(\boldsymbol{\pi}_k)\boldsymbol{Z}^T)}_{R_c(\boldsymbol{Z}, \boldsymbol{\Pi})} \quad (1)$$

Here, the expansion term $R(\boldsymbol{Z})$ measures the volume of the overall feature space, thereby promoting inter-class feature separation. In contrast, the compression term $R_c(\boldsymbol{Z}, \boldsymbol{\Pi})$ measures the sum of volumes of feature subspaces from each of the $K$ classes encoded by $\boldsymbol{\Pi}$ and minimizing it encourages intra-class compactness. To maximize Eq. (1), the operators in each layer of ReduNet, roughly in the form of $(\boldsymbol{I} + \boldsymbol{Z}\boldsymbol{Z}^T)^{-1}$, are derived from the gradient ascent of MCR$^2$, and perform expansion and compression with equal weights. While ReduNet offers a principled design, its dependence on full training features and $\mathcal{O}(n^3)$ matrix inversions limits its scalability to large datasets.

Rather than constructing each layer's operators directly from the features and compressing them into subspaces formed by the features themselves, CRATE compresses features into $K$ learnable subspaces $\boldsymbol{U}_{[K]}$ at each layer, enabling efficient batch-wise training on large datasets via backpropagation. Specifically, CRATE proposes a modified compression term $R_c(\boldsymbol{Z}|\boldsymbol{U}_{[K]})$ by incorporating $K$ trainable subspace matrices concatenated as $\boldsymbol{U}_{[K]} = [\boldsymbol{U}_1, \dots, \boldsymbol{U}_K] \in \mathbb{R}^{d \times Kp}$. Each $\boldsymbol{U}_i \in \boldsymbol{R}^{d \times p}$ (with $p < d$) spans the basis of the $i$-th low-dimensional subspace. The modified compression term is defined as follows:

$$R_c(\boldsymbol{Z}|\boldsymbol{U}_{[K]}) = \frac{1}{2} \sum_{k=1}^{K} \log \det(\boldsymbol{I} + \frac{p}{n\epsilon^2}(\boldsymbol{U}_k^T\boldsymbol{Z})^T(\boldsymbol{U}_k^T\boldsymbol{Z})) \tag{2}$$

Here, $R_c(\boldsymbol{Z}|\boldsymbol{U}_{[K]})$ measures the compactness of representations $\boldsymbol{U}_k^T\boldsymbol{Z}$ within each low-dimensional subspace. CRATE approximates the gradient descent step of $R_c(\boldsymbol{Z}|\boldsymbol{U}_{[K]})$ to avoid the expensive matrix inverse, deriving the MSSA module of each layer:

$$\text{MSSA}(\boldsymbol{Z}|\boldsymbol{U}_{[K]}) = [\boldsymbol{U}_1, \dots, \boldsymbol{U}_K] \begin{bmatrix} \boldsymbol{U}_1^T\boldsymbol{Z}\text{softmax}((\boldsymbol{U}_1^T\boldsymbol{Z})^T(\boldsymbol{U}_1^T\boldsymbol{Z})) \\ \vdots \\ \boldsymbol{U}_K^T\boldsymbol{Z}\text{softmax}((\boldsymbol{U}_K^T\boldsymbol{Z})^T(\boldsymbol{U}_K^T\boldsymbol{Z})) \end{bmatrix} \tag{3}$$

However, MSSA violates the compression goal as it actually performs the opposite operation, i.e., increasing compression term (Hu et al., 2024). The reason behind this is that MSSA is the result of a second-order gradient approximation of $R_c(\boldsymbol{Z}|\boldsymbol{U}_{[K]})$, but this approximation omits the first-order term. Furthermore, MSSA suffers from inefficiency as it requires the computation of pairwise similarities $(\boldsymbol{U}_k^T\boldsymbol{Z})^T(\boldsymbol{U}_k^T\boldsymbol{Z})$ across all input tokens, leading to quadratic complexity.

ToST further reduces computational complexity by introducing an upper-bound variant of Eq. (2), which relies only on the diagonal elements of the correlation matrix $(\boldsymbol{U}_k^T\boldsymbol{Z}\text{Diag}(\boldsymbol{\pi}_k)\boldsymbol{Z}^T\boldsymbol{U}_k)$. By performing a gradient descent step on this upper-bound variant, ToST derives the TSSA module. However, since it only leverages the diagonal elements, it neglects correlations among dimensions, thereby failing to effectively capture the fine-grained structural information for feature updates (See Appendix A.1 for detailed analysis).

Moreover, in both models, each layer lacks an explicit expansion module: MSSA is followed by a module primarily promoting feature sparsity, while TSSA is followed by a simple MLP layer. This omission undermines the principled approach these models aim to provide for architecture design. In addition, both models simplify the implementation: specifically, MSSA replaces the matrices $[\boldsymbol{U}_1, \dots, \boldsymbol{U}_K]$ in Eq. (3) with a single trainable parameter matrix $\boldsymbol{W}$, which is inconsistent with the use of $\boldsymbol{U}_{[K]}$ elsewhere in MSSA, while TSSA adopts a similar simplification. These design choices further compromise the intended principled structure.

## 3 ECATTENTION

Our goal remains to expand all features ($R(\boldsymbol{Z})$ in Eq. (1)) and compress the representations within each subspace ($R_c(\boldsymbol{Z}|\boldsymbol{U}_{[K]})$ in Eq. (2)). Unlike prior operators that directly use or approximate the gradient of the objective function, we exploit the geometric interpretation of the gradient of $\log \det(\boldsymbol{I} + \boldsymbol{Z}\boldsymbol{Z}^T)$, a term that appears repeatedly in MCR$^2$ and its variants, to redesign the expansion and compression modules. Consider a PSD matrix $\boldsymbol{M} = \boldsymbol{Z}\boldsymbol{Z}^T \in \text{PSD}(d)$, $\log \det(\boldsymbol{I} + \boldsymbol{M}) = \sum_{i=1}^{d} \log(1 + \lambda_i(\boldsymbol{M}))$ where $\lambda_i(\boldsymbol{M})$ is the $i$-th largest eigenvalue of $\boldsymbol{M}$. To increase the value of $\log \det(\cdot)$, we can either increase the zero eigenvalues (corresponding to the null space $\text{null}(\boldsymbol{Z}^T)$) or enhance the non-zero eigenvalues (corresponding to the column space

range($\boldsymbol{Z}$)). To achieve **effective feature expansion**, *the eigenvalues associated with the null space should be enlarged to increase the dimensionality of the column space.* In this way, the rank of the matrix $\boldsymbol{M}$ will increase, meaning that it spans a larger space and could promote feature separation. In contrast, **effective feature compression** should be realized by driving certain non-zero eigenvalues toward zero to reduce dimensionality of the column space.

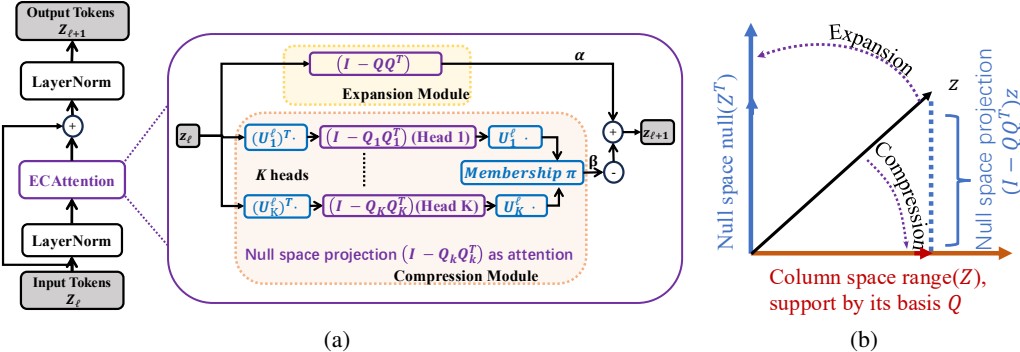

$$(a) \qquad\qquad\qquad\qquad (b)$$

Figure 1: (a) One layer of ECA Transformer. (b) Illustration of expansion and compression.

## 3.1 OVERVIEW OF ECATTENTION

Based on the above analysis, an effective operation involves two steps: (1) identifying the column space and null space. (2) Increasing the dimensionality of column space to expand feature space or reducing it to compress feature space. As illustrated in Figure 1(b), adding or subtracting the projection of a feature vector $\boldsymbol{z}$ onto the null space moves $\boldsymbol{z}$ toward the null space or column space, thereby realizing feature expansion or compression. Let $\boldsymbol{Q}$ be the orthonormal basis of the identified column space range($\boldsymbol{Z}$). Then the projection operation is straightforward. The projection of a vector $\boldsymbol{z}$ onto the column space is simply $\boldsymbol{Q}\boldsymbol{Q}^T\boldsymbol{z}$, whereas the projection onto the null space is $(\boldsymbol{I} - \boldsymbol{Q}\boldsymbol{Q}^T)\boldsymbol{z}$ (Fessler & Nadakuditi, 2024). From this geometric insight, we propose **ECAttention** to achieve effective feature compression and expansion at each layer, respectively. This yields the **ECA** transformer architecture, where each layer's feature update is defined as follows.

$$\boldsymbol{z}_{\ell+1} = \boldsymbol{z}_\ell + \underbrace{\alpha(\boldsymbol{I} - \boldsymbol{Q}\boldsymbol{Q}^T)\boldsymbol{z}_\ell}_{\text{Expansion}} - \underbrace{\beta\sum_{k=1}^{K}\pi_k\boldsymbol{U}_k(\boldsymbol{I} - \boldsymbol{Q}_k\boldsymbol{Q}_k^T)\boldsymbol{U}_k^T\boldsymbol{z}_\ell}_{\text{Compression}} \qquad (4)$$

Here, $\boldsymbol{Q}_k$ denotes the orthonormal basis of $\boldsymbol{U}_k^T\boldsymbol{Z}$. For clarity and brevity of presentation, we suppress the layer index $\ell$ on all layer-specific parameter matrices ($\boldsymbol{Q}_{(\ell)}$, $\boldsymbol{U}_{(\ell)}$, etc.) unless explicitly required. Figure 1(a) gives an overview of a layer of ECA. The resulting architecture remains remarkably simple. At each layer, every feature is transformed by ECAttention and subsequently added back to the input via a residual connection. The expansion module aims to enlarge the coding rate of all features $R(\boldsymbol{Z})$ in Eq. (1), while the compression module seeks to decrease the coding rate of intra-group features $R_c(\boldsymbol{Z}|\boldsymbol{U}_{[K]})$ in Eq. (2). The trainable parameters $\alpha$ and $\beta$ are optimized by backpropagation during training to learn the strength of expansion and compression at each layer.

Unlike the expansion module that directly updates features via null space projection, trainable parameter matrices $\boldsymbol{U}_{[K]}$ in the compression module can be viewed as $K$ bases. Each token feature $\boldsymbol{z}$ is first projected onto the basis $\boldsymbol{U}_k$ by multiplying by $\boldsymbol{U}_k^T$, then updated according to the null space projection $(\boldsymbol{I} - \boldsymbol{Q}_k\boldsymbol{Q}_k^T)$, and finally projected back to the standard basis by multiplying by $\boldsymbol{U}_k$. The membership $\boldsymbol{\pi} = [\pi_1, \ldots, \pi_K]$, computed from the similarity between $\boldsymbol{z}$ and bases $\boldsymbol{U}_{[K]}$, gives the probability of the token belonging to each group, guiding its update toward the corresponding group.

## 3.2 CONSTRUCT EXPANSION & COMPRESSION MODULES BASED ON ORTHONORMAL BASIS

The expansion module is designed to increase the coding rate $R(\boldsymbol{Z})$ layer by layer. This is achieved in two steps: (1) project each feature $\boldsymbol{z} \in \boldsymbol{Z}$ onto the null space null($\boldsymbol{Z}^T$), and (2) add the projection

to each feature so that it moves towards the null space of the overall features. Hence, after obtaining the orthonormal basis $\boldsymbol{Q} \in \mathbb{R}^{d \times Kr}$ of $\boldsymbol{Z}$, the expansion update can be denoted as:

$$\boldsymbol{z}_{\ell+1} = \boldsymbol{z}_\ell + (\boldsymbol{I} - \boldsymbol{Q}\boldsymbol{Q}^T)\boldsymbol{z}_\ell \in \mathbb{R}^d \tag{5}$$

Here, $Kr \leq d$, where $K$ is the number of heads, which denotes the number of subspaces. $r$ is the rank of each subspace. Hence, $Kr$ denotes the dimension of the overall feature column space.

When it comes to the compression module, the situation is different as the object to be compressed in Eq. (2) is the *code* (i.e., a low-dimensional representation) of the token feature $\boldsymbol{\alpha}_k = \boldsymbol{U}_k^T \boldsymbol{z}$ rather than the feature $\boldsymbol{z}$ (see Appendix A.2 for a formal description). The *codes* of all features on subspace $\boldsymbol{U}_k$ can be denoted as $\boldsymbol{A}_k = \boldsymbol{U}_k^T \boldsymbol{Z} \in \mathbb{R}^{p \times n}$ and its column space is $\boldsymbol{Q}_k \in \mathbb{R}^{p \times r}$. Hence, for a *code* $\boldsymbol{\alpha}_k$ of token feature, the projection onto the null space $\text{null}(\boldsymbol{A}_k^T)$ and column space $\text{range}(\boldsymbol{A}_k)$ are $(\boldsymbol{I} - \boldsymbol{Q}_k\boldsymbol{Q}_k^T)\boldsymbol{\alpha}_k$ and $\boldsymbol{Q}_k\boldsymbol{Q}_k^T\boldsymbol{\alpha}_k$, respectively. Thus, by subtracting the null space projection, we can achieve the goal of compressing *code* $\boldsymbol{\alpha}_k$ toward their corresponding subspace.

$$\boldsymbol{\alpha}_k^{\ell+1} = \boldsymbol{\alpha}_k^\ell - (\boldsymbol{I} - \boldsymbol{Q}_k\boldsymbol{Q}_k^T)\boldsymbol{\alpha}_k^\ell \in \mathbb{R}^p \tag{6}$$

The projection onto the column spaces $\{\boldsymbol{Q}_k\boldsymbol{Q}_k^T\boldsymbol{\alpha}_k^\ell\}_{k=1}^K$ can be converted to a distribution of membership $\boldsymbol{\pi}$ as follows:

$$\boldsymbol{\pi} = \text{softmax}\left(\frac{1}{2\eta}\begin{bmatrix} ||\boldsymbol{Q}_1\boldsymbol{Q}_1^T\boldsymbol{\alpha}_1^\ell||^2 \\ \vdots \\ ||\boldsymbol{Q}_K\boldsymbol{Q}_K^T\boldsymbol{\alpha}_K^\ell||^2 \end{bmatrix}\right) = [\pi_1, \dots, \pi_K] \in \mathbb{R}^K \tag{7}$$

This quantifies the probability that a token belongs to each subspace, which can be used as weights to guide each feature to update towards its corresponding subspace. Finally, we obtain the following update formula for feature compression (detailed derivation can be found in the Appendix A.2):

$$\boldsymbol{z}_{\ell+1} \approx \boldsymbol{z}_\ell - \sum_{k=1}^K \pi_k \boldsymbol{U}_k(\boldsymbol{I} - \boldsymbol{Q}_k\boldsymbol{Q}_k^T)\boldsymbol{U}_k^T\boldsymbol{z}_\ell \tag{8}$$

### 3.3 HOW TO OBTAIN THE BASIS OF COLUMN SPACE EFFECTIVELY?

One key operation in ECAttention is to identify the column space of the feature matrix, which is typically considered computationally intensive. In this work, we choose to leverage a randomization method to capture column space efficiently (Halko et al., 2011). Once the basis $\boldsymbol{Q}$ of the column space is obtained, the projection onto the null space is given by $(\boldsymbol{I} - \boldsymbol{Q}\boldsymbol{Q}^T)$. Specifically, suppose that we seek the column space of feature matrix $\boldsymbol{Z} = [\boldsymbol{z}_1 \cdots \boldsymbol{z}_n] \in \mathbb{R}^{d \times n}$ with a specified rank $r$. A random matrix $\boldsymbol{\Omega} = [\boldsymbol{\omega}_1 \cdots \boldsymbol{\omega}_r] \in \mathbb{R}^{n \times r}$ can be multiplied with $\boldsymbol{Z}$: $\boldsymbol{Y} = [\boldsymbol{Z}\boldsymbol{\omega}_1 \cdots \boldsymbol{Z}\boldsymbol{\omega}_r] = [\boldsymbol{y}_1 \cdots \boldsymbol{y}_r] \in \mathbb{R}^{d \times r}$. Due to the randomness, the random vectors $[\boldsymbol{\omega}_1 \cdots \boldsymbol{\omega}_r]$ form a linearly independent set with high probability. The resulting linear combinations $[\boldsymbol{y}_1 \cdots \boldsymbol{y}_r]$ lie in the column space of $\boldsymbol{Z}$ and are also linearly independent, thereby spanning the column space of $\boldsymbol{Z}$. This also plays a role in dimensionality reduction, thereby resulting in a reduced-dimension compression space. Subsequently, given the higher efficiency of Cholesky decomposition compared to QR decomposition, we employ Cholesky decomposition to further process the Gram matrix $\boldsymbol{Y}^T\boldsymbol{Y}$ to obtain an orthonormal basis $\boldsymbol{Q}$ (see Appendix A.3 for details).

**Complexity analysis.** Since our expansion module is aligned with the MLP module in ToST, it is appropriate to compare only the proposed compression module with the TSSA module. The time complexity of the compression module is almost equivalent to that of the TSSA module ($\mathcal{O}(pn)$), which is a **linear-time** attention, except that our module includes an additional cost for Cholesky decomposition ($\mathcal{O}(\frac{1}{3}r^3)$) that aims to capture more precise information for feature updates. Since the rank $r$ of random matrix $\boldsymbol{\Omega}$ is predefined as constant, the time cost of the Cholesky decomposition remains stable. Hence, as Figure 2(a) shows, although ECA has slightly higher time complexity when the number of tokens is small, as the number of tokens increases, the time complexity of ECAttention becomes **almost identical to or even lower** than that of ToST. After removing the Cholesky decomposition step (ECA w/o Cholesky), our model shows lower time complexity, which confirms that the source of additional time cost comes from the Cholesky decomposition. In contrast, the attention operators of CRATE exhibit quadratic time complexity of $\mathcal{O}(pn^2)$, which is the same order

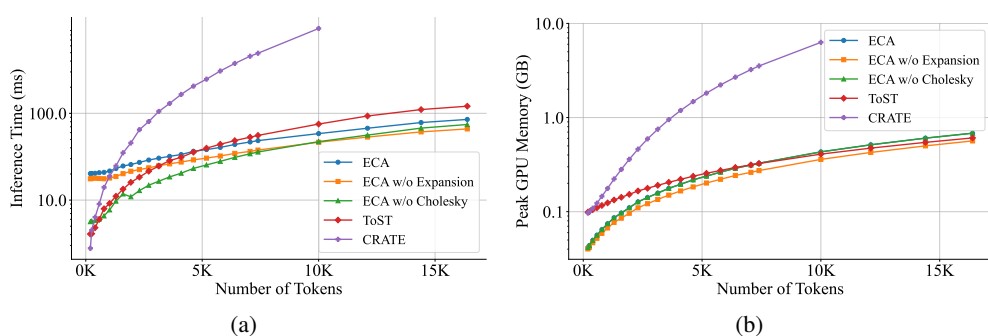

Figure 2: Inference time (*Left*) and memory usage (*Right*) versus the number $n$ of input tokens. All models are measured with 12 layers, $d = 384$, and $K = 8$. The y-axes are in log-scale.

of complexity as that of the classical Vision Transformer (ViT) (Wu et al., 2025). Besides, Figure 2(b) suggests that our model exhibits linear memory usage as the number of tokens $n$ increases. The step-by-step complexity analysis are shown in Appendix A.4.

The computational overhead of Cholesky decomposition can be further reduced by decreasing the value of $r$, since the self-adaptive capabilities of expansion and compression modules allow small-scale ECA models to retain their performance even at low $r$. Detailed experimental analysis is provided in Section 4.3.

## 4 EXPERIMENTS

In this section, we conduct experiments on both toy data and real-world datasets to verify and study the properties and performance of our proposed ECA transformer. We adopt a straightforward implementation that strictly follows our formulation. Therefore, demonstrating that the current implementation of ECA outperforms existing highly engineered architectures is not the goal of this work. Rather, our empirical studies aim to provide answers and evidence for the following questions:

1. Do the expansion and compression modules achieve effective operation?

2. Is compression via null-space subtraction more stable than column-space projection?

3. Does the ECA transformer exhibit better interpretability while maintaining task performance?

4. Does the ECA transformer exhibit self-adaptive behavior?

Our framework uses ECAttention as the backbone, while other components, such as tokenization and positional encoding, largely follow the ToST implementation. To address the above three questions, we perform experiments across models of varying scales by adjusting the number of attention heads $K$, the token dimension $d$, and the number of layers $L$. Due to space limitations, the details on practical implementation and configurations are provided in Appendices B.1 and B.2. Additional ablation studies and the pseudocode of ECAttention are provided in Appendices C and D.

**Datasets and training configuration.** We pre-train the ECA models on ImageNet-1K dataset (Deng et al., 2009; Beyer et al., 2020). The pre-trained models are subsequently adapted through fine-tuning on downstream tasks using CIFAR-10/100 (Krizhevsky et al., 2009), Oxford Flowers (Nilsback & Zisserman, 2008), and Oxford-IIT-Pets (Parkhi et al., 2012) datasets. We set the initial values of learnable parameters to $\alpha = 0.1$ and $\beta = 0.1$ for each layer, with $r = 20$ and $500$ epochs for pre-training. More training details are provided in Appendix B.4.

### 4.1 VERIFICATION ON TOY DATA

**Effective expansion and compression** We begin by using synthetic toy data to validate the effectiveness of compression and expansion modules. Specifically, we assume that each

feature $\boldsymbol{z}$ in $\boldsymbol{Z} \in \mathbb{R}^{d \times n}$ is generated from one of $K = 6$ mutually orthogonal subspaces $\boldsymbol{U}_{[K]}$, where each subspace is supported by an orthonormal basis $\boldsymbol{U}_k \in \mathbb{R}^{d \times p}$. We set $d = 384$ and $p = 64$, and construct each sample as $\boldsymbol{z} = \boldsymbol{U}_k \boldsymbol{\alpha}$ by activating only the first 20 dimensions of $\boldsymbol{U}_k$ through $\boldsymbol{\alpha}$. In total, we generate $n = 1000$ samples. The resulting dataset has an overall rank of 120, with each subspace being 20-dimensional.

We apply the compression and expansion operations to this data separately. The layer-wise coding rates are computed according to the expansion term $R(\boldsymbol{Z})$ in Eq. (1) and the modified compression term $R_c(\boldsymbol{Z}|\boldsymbol{U}_{[K]})$ in Eq. (2). As shown in Figure 3(a), the proposed expansion and compression modules successfully achieve their intended design objectives, with $R(\boldsymbol{Z})$ be-

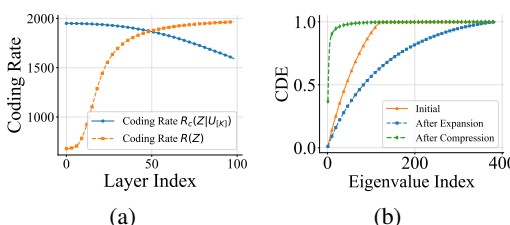

Figure 3: (a) Layer-wise changes in the coding rate; (b) Changes in the cumulative distribution of eigenvalues (CDE) of $\boldsymbol{Z}\boldsymbol{Z}^T$.

ing incrementally expanded and $R_c(\boldsymbol{Z}|\boldsymbol{U}_{[K]})$ being gradually reduced across layers. Figure 3(b) shows the cumulative distribution of eigenvalues, which reflects how many dimensions carry informative content out of the 384 dimensions. Initially, the dataset has a rank of 120, meaning that the first 120 dimensions capture all the energy (i.e., correspond to nonzero eigenvalues). As shown in Figure 3(b), the proposed modules achieve effective expansion and compression: expansion is realized by increasing the eigenvalues associated with the null space, thereby enlarging the effective dimensions, while compression narrows the dimensions of the column space.

**Compression by subtracting null space projection allows effective optimization** We choose to subtract the null space projection (i.e., $\boldsymbol{I} - \boldsymbol{Q}_k \boldsymbol{Q}_k^T$) rather than directly projecting onto the column space (i.e., $\boldsymbol{Q}_k \boldsymbol{Q}_k^T$) for feature compression in each layer. As shown in Figure 4, compression by subtracting null space projection enables effective multi-epoch optimization. The reason is that compression via subtracting the null-space projection yields a skip-connection–like update, i.e., $\boldsymbol{z}_{\ell+1} = \boldsymbol{z}_\ell - (\text{null-space projection})$, which is more favorable for multi-epoch and layer-wise optimization. In contrast, directly projecting onto the column space results in a structure like $\boldsymbol{z}_{\ell+1} = (\text{column-space projection})$. This structure blocks the input information of each layer from flowing to subsequent layers, which undermines both layer-

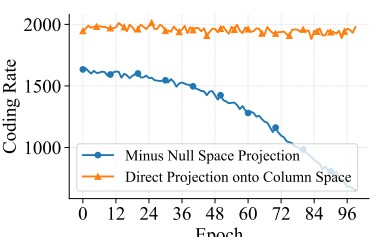

Figure 4: Compression via column-space projection vs. subtracting null-space projection.

wise optimization and multi-epoch training. (See Appendix C.4 for more details).

### 4.2 EXPERIMENTS ON REAL-WORLD VISUAL DATA

**Training & fine-tuning results.** We primarily compare ECA with ToST. The results for DeiT (Touvron et al., 2021) (a data-efficient ViT variant) and CRATE are directly reported from their original papers, as both methods are computationally prohibitive to reproduce in our setting. As shown in Figure 5, ECA demonstrates better performance than CRATE, and is also comparable to ToST when the parameter scale is small. However, at larger parameter scales, ECA falls slightly behind ToST. There are two factors affecting the comparison in this experiment: one is the unprincipled MLP module in ToST, and the other is the setting of the hyperparameter $r$ in our model. Specifically, we found that nearly two-thirds of ToST's parameters belong to its MLP module, indicating that ToST's superior performance mainly comes from the MLP rather than its compression component. On the other hand, in our current large-scale experiments, we set the hyperparameter $r = 20$. This parameter controls how many dimensions each layer captures for feature updates, and a larger $r$ generally leads to higher accuracy. Due to our limited computational resources, conducting large-scale comparative studies on ImageNet-1K would take several months. Hence, results with larger $r$ on large-scale datasets are not available in the current study. Nevertheless, we will provide a more detailed comparative analysis in subsequent experiments on small-scale validation datasets(see Section 4.4).

Table 1 presents the top-1 accuracy of ECA on the ImageNet-1K dataset and fine-tuning accuracy across several smaller datasets. For CIFAR-10/100, we report fine-tuning results over 200 epochs, while for Oxford Flowers-102 and Oxford-IIIT-Pets, we report results over 30 epochs. Overall, ECA achieves results comparable to ToST. Specifically, when comparing ECA-S with ToST-T, ECA-S delivers better performance, and we further observe that ECA-S converges faster during fine-tuning on Oxford Flowers-102. In addition, when comparing ECA-M with ToST-S, ECA-M attains comparable results while using only two-thirds of the parameters. ECA-B+ fails to achieve the substantial accuracy gains suggested by its increased parameter count, as setting $r = 20$ remains suboptimal for larger models. Despite this, ECA still exhibits trends that align with the scaling law.

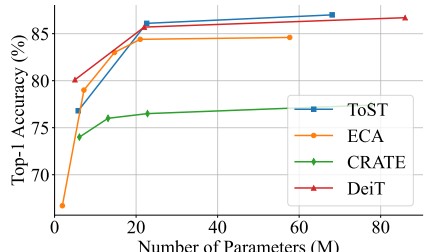

Figure 5: Top-1 accuracy of four models on ImageNet-Real across parameter scales.

Table 1: Top 1 accuracy of ECA across datasets with different model sizes. For ImageNet and ImageNet ReaL, we directly evaluate the top-1 accuracy. For other datasets, we pre-train the model on ImageNet and fine-tune it. Due to space limits, DeiT results are provided in Appendix B.3

| Datasets | ECA-S | ECA-M | ECA-B+ | ToST-T | ToST-S | ToST-M | CRATE-B | CRATE-L |
|---|---|---|---|---|---|---|---|---|
| # parameters | 7.2M | 14.7M | 57.7M | 5.8M | 22.6M | 68.1M | 22.8M | 77.6M |
| ImageNet | 67.9 | 73.1 | 75.6 | 64.9 | 77.5 | 79.6 | 70.8 | 71.3 |
| ImageNet ReaL | 79.0 | 83.0 | 84.6 | 76.8 | 86.1 | 87.0 | 76.5 | 77.4 |
| CIFAR10 | 94.1 | 95.2 | 96.2 | 94.7 | 97.4 | 98.0 | 96.8 | 97.2 |
| CIFAR100 | 77.3 | 81.2 | 82.2 | 77.2 | 85.4 | 86.1 | 82.7 | 83.6 |
| Oxford Flowers-102 | 75.5 | 83.4 | 99.5 | 49.1 | 95.4 | 98.8 | 88.7 | 88.3 |
| Oxford-IIIT-Pets | 92.6 | 95.7 | 99.2 | 86.0 | 98.6 | 99.5 | 85.3 | 87.4 |

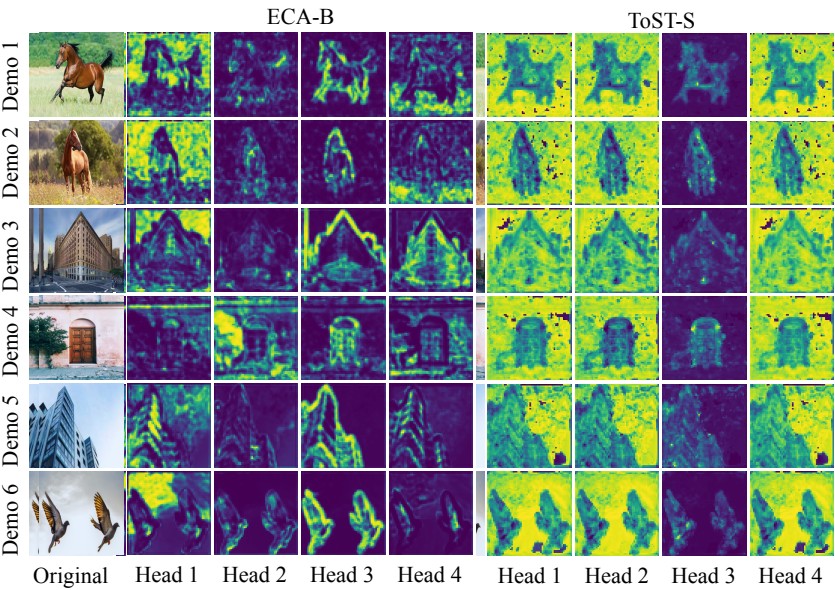

Figure 6: **Membership distribution $\pi$ of ECA-B (*left*) vs. ToST-S (*right*). Different heads in ECA-B capture distinct parts, enabling more fine-grained structural modeling.** Visualization of head-wise membership $\pi_k$ (reshaped to $\sqrt{N} \times \sqrt{N}$ for $N$ tokens). Results are shown for layer 3 in ECA-B and layer 9 in ToST-S.

**Improved interpretability in membership visualization.** At each layer, Eq. (7) estimates the soft memberships $\pi$ of each token, which provides the probability that a token will be assigned to the $k$-th group or subspace. Hence, we could obtain the $K$ clusters of $N$ tokens. As Figure 6 shows, following the design principle of compressing tokens into different subspaces, the ECA framework automatically learns object segmentation without complex self-supervised learning recipes or segmentation-related annotations. In particular, heads 3 and 4 in ECA-B not only identify objects, but also perform more **fine-grained segmentation**. For example, in the building image (demo 5), head 3 captures the overall framework, while head 4 focuses on the detailed structure of individual floors. Similarly, in demo 4, head 2 identifies the tree while head 3 attends to the door. In contrast, ToST-S heads mainly focus on coarse object–background separation. Note that bright yellow indicates the attended regions, while only one head (head 3) of ToST focuses on the object.

### 4.3 HYPERPARAMETER ANALYSIS

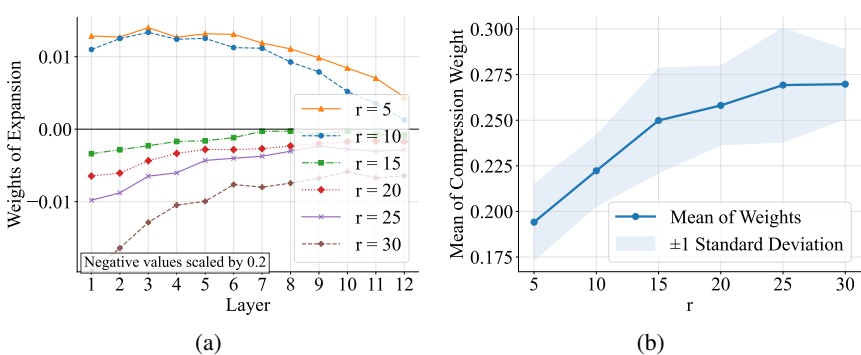

(a)  (b)

Figure 7: (*Left*) Expansion weights $\alpha$ across layers for different $r$ values. (*Right*) Mean of compression weights $\beta$ across layers for different $r$ values.

We introduce two learnable weight parameters, $\alpha$ and $\beta$, for the expansion module and compression module, respectively, in Eq. (4), and one hyperparameter $r$ for the random matrix $\Omega$. As mentioned in Section 3.3, $r$ is a predefined rank that controls the dimensionality of the compressed subspace by determining the size of the random matrix $\Omega$. Therefore, to investigate the effect of the hyperparameter $r$, we evaluate our approach on the CIFAR-10 dataset with two model variants: ECA-S (7.2M parameters) and ECA-B (21M parameters).

**Self-adaptive capabilities of expansion and compression.** Figure 7 illustrates the evolution of the learnable weight parameters $\alpha$ and $\beta$ during the training of ECA-S on CIFAR-10, initialized at 0.1 and trained for 400 epochs. As Figure 7(a) shows, when $r \leq 10$, the weights of the expansion module $\alpha$ in the early layers are higher than those in the later layers, which aligns with previous findings (Yu et al., 2024). Specifically, it indicates that features should be expanded during early layers, thereby promoting inter-group separation and facilitating compression of group features into their corresponding subspaces in deeper layers. Nevertheless, when $r \geq 15$, the expansion module weights ultimately converge to negative values, indicating that when we set $r$ to compress features into larger subspaces, the expansion module tends to compress the entire feature space rather than expanding it. For the compression module, Figure 7(b) presents the variation in the mean values of the compression module weights $\beta$ across layers as $r$ increases. As observed, when the predefined compression space becomes larger, the mean weight values also rise, suggesting that the network increasingly favors feature compression.

**Enforcing weight positivity to regulate expansion–compression dynamics.** As discussed above, directly optimizing the weight parameters of each layer may cause the expansion module to perform compression instead. Besides, as shown in Figure 8(a), the compression module weights of some layers are zero. This indicates that without imposing any constraint on the weights, certain layers may not be activated at all. To address this, we enforce the weight parameters to be positive by applying the softplus function(Dugas et al., 2000). Enforcing positivity (ECA-B + softplus) ensures

that all layers remain active and also reduces the fluctuation of the weights. In addition, as shown in Figure 8(b), this trick makes the effect of tuning the hyperparameter $r$ more stable: a larger value of $r$ results in higher accuracy. Considering both computational cost and accuracy, we recommend setting $r = 20$.

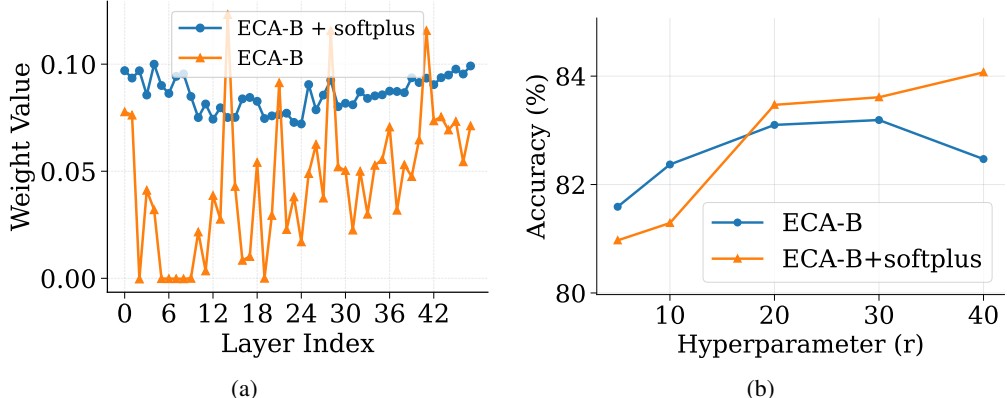

(a)                                                                (b)

Figure 8: (a) The weight distribution of compression modules. We train the ECA-B on CIFAR-10 with $r = 5$. (b) Softplus trick stabilizes the effect of tuning the hyperparameter $r$.

### 4.4 A FAIRER COMPARISON WITH ToST

We take ToST-S (22.6M) as an example and find that nearly two-thirds of its parameters come from the MLP module. Its superior performance is therefore mainly attributed to the MLP module rather than the proposed compression module. When we remove the MLP in ToST-S, its parameter count drops to 8M. Accordingly, we compare it with our ECA-B (20M) and ECA-S (6M) models, both equipped with the weight positivity constraint. As shown in Figure 9, after removing MLP module, ToST-S-w/o MLP exhibits the lowest accuracy of 81.16%. In contrast, ECA-S, with an even smaller parameter size (6M), surpasses ToST-S-w/o MLP. The figure also illustrates that a larger $r$ or more parameters indeed leads to corresponding improvements in accuracy. Due to space limitations, the results on CIFAR-100 are provided in Appendix C.2.

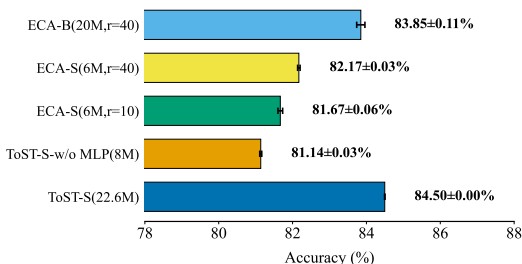

Figure 9: A fairer comparison with ToST on CIFAR-10 after 100 epochs training.

## 5 CONCLUSION

In this work, we design a novel principled attention to achieve effective feature compression and expansion with low complexity, guided by the geometric insight of the gradient of $\text{MCR}^2$ objective. We leverage randomization combined with Cholesky decomposition to efficiently obtain the column space of the feature matrix and its basis, enabling the construction of ECAttention, which includes compression and expansion modules. The resulting ECA transformer demonstrates higher interpretability and achieves fine-grained modeling of data structure for token feature updates with low computational complexity. Additionally, ECA exhibits self-adaptive capability in learning the strength of expansion and compression across layers. Compared with ToST, CRATE, our framework adheres more closely to the white-box design principle and achieves better or comparable performance.

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

## A   THEORETICAL ANALYSIS

### A.1   ANALYSIS OF TSSA

For a matrix $\boldsymbol{M} \in \mathbb{R}^{m \times n}$, we denote by $\boldsymbol{M}^{\odot 2} \in \mathbb{R}^{m \times n}$ the element-wise square of $\boldsymbol{M}$. For a function $f : \mathbb{R} \to \mathbb{R}$, let $f[\boldsymbol{v}] \in \mathbb{R}^n$ be the element-wise application of $f$ to the entries of $\boldsymbol{v}$.

TSSA uses second-moment statistics for feature update as shown in Eq. (9).

$$\mathrm{D}(\boldsymbol{Z}, \boldsymbol{\pi}_k | \boldsymbol{U}_k) = \mathrm{Diag}\left(\nabla f\left[(\boldsymbol{U}_k^T \boldsymbol{Z})^{\odot 2} \frac{\boldsymbol{\pi}_k}{\langle \boldsymbol{\pi}_k, \mathbf{1} \rangle}\right]\right) \tag{9}$$

Here, $f(x) = \log(1 + x)$. The $\nabla f[\cdot]$ is applied to each element of the vector in the bracket. To facilitate understanding, we demonstrate TSSA's operation through a simple example involving two samples with two-dimensional features $\boldsymbol{Z} = [\boldsymbol{z}_1, \boldsymbol{z}_2] \in \mathbb{R}^{2 \times 2}$ and membership vector $\boldsymbol{\pi}_k = [\pi_1, \pi_2]^T \in \mathbb{R}^{2 \times 1}$ represents the probability that each sample belongs to $k$-th group.

$$\mathrm{Diag}(\boldsymbol{Z}^{\odot 2} \boldsymbol{\pi}_k) = \mathrm{Diag}\left(\begin{bmatrix} z_{11}^2 & z_{21}^2 \\ z_{12}^2 & z_{22}^2 \end{bmatrix} \begin{bmatrix} \pi_1 \\ \pi_2 \end{bmatrix}\right) = \mathrm{Diag}\left(\begin{bmatrix} z_{11}^2 \pi_1 + z_{21}^2 \pi_2 \\ z_{12}^2 \pi_1 + z_{22}^2 \pi_2 \end{bmatrix}\right) \tag{10}$$

This demonstrates that the $i$-th diagonal element of TSSA captures the sum of squared $i$-th dimensional components across samples in the $k$-th group, thereby providing statistical information that informs feature update mechanisms. Essentially, this is equivalent to the "weighted" diagonal elements of the covariance matrix $\boldsymbol{Z} \mathrm{Diag}(\boldsymbol{\pi}_k) \boldsymbol{Z}^T$, thus ignoring the correlations between different dimensions. This can be formalized as follows:

$$\boldsymbol{Z} \mathrm{Diag}(\boldsymbol{\pi}_k) \boldsymbol{Z}^T = \begin{bmatrix} z_{11}^2 \pi_1 + z_{21}^2 \pi_2 & z_{11} z_{12} \pi_1 + z_{21} z_{22} \pi_2 \\ z_{12} z_{11} \pi_1 + z_{22} z_{21} \pi_2 & z_{12}^2 \pi_1 + z_{22}^2 \pi_2 \end{bmatrix} \tag{11}$$

We also give a toy data example to show that such simple statistics are inadequate for capturing

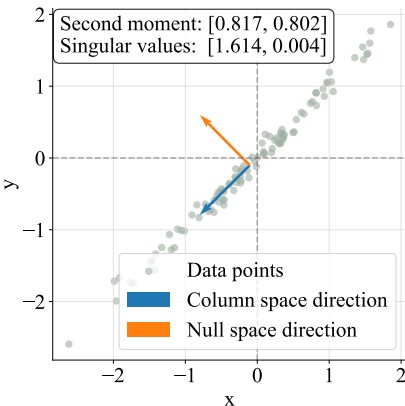

Figure 10: Second-moment statistic vs Singular values

fine-grained structures that can effectively guide feature compression. Consider that 100 data points $(x, y)$ are generated from the linear model $y = x + \epsilon$, where $\epsilon \sim \mathcal{N}(0, 0.1)$ is a Gaussian noise.

Since the linear dependence between variables $x$ and $y$, the rank of the generated data matrix should be 1. As Figure 10 shows, the first singular value obtained by singular value decomposition (SVD) is significantly larger than the second, indicating that the true dimensionality of the data is one-dimensional. In contrast, the second moment statistics do not reveal a clear distinction between the two dimensions of the data.

## A.2 PROOF OF THE COMPRESSION FORMULA

In the main text, we noted that the compression module operates not directly on the token feature $\boldsymbol{z}$, but rather on its code $\boldsymbol{\alpha}_k = \boldsymbol{U}_k^T \boldsymbol{z}$ with respect to the basis $\boldsymbol{U}_k$. Here, we provide a formal mathematical description of the assumption behind this. Specifically, the definition of $R_c(\boldsymbol{Z} \mid \boldsymbol{U}_{[K]})$ is grounded in the **union of $K$ low-dimensional subspace models**, involving $K$ orthonormal basis matrices. Assume that $\boldsymbol{U}_{[K]} = [\boldsymbol{U}_1, \ldots, \boldsymbol{U}_K] \in \mathbb{R}^{d \times Kp}$ is a set of orthonormal basis matrices for $K$ subspaces with $\boldsymbol{U}_k \in \mathbb{R}^{d \times p}$. We say that a token feature $\boldsymbol{z} = \boldsymbol{U}_k \boldsymbol{\alpha} \in \mathbb{R}^d$ lies on the union of subspaces supported by $\boldsymbol{U}_{[K]}$. From an information theory perspective, $\boldsymbol{U}_{[K]}$ can be viewed as codebooks and the vectors $\boldsymbol{\alpha} = \boldsymbol{U}_k^T \boldsymbol{z} \in \mathbb{R}^p$ can be the codes of the token feature $\boldsymbol{z}$ with respect to the codebook $\boldsymbol{U}_k$(Yu et al., 2023; Xu et al., 2025).

The compression module is derived from the update of *codes* $\boldsymbol{\alpha}_k$ as shown in Eq. (6). Given a token feature $\boldsymbol{z}$ and its *code* $\boldsymbol{\alpha}_k$ in $k$-th subspace supported by $\boldsymbol{U}_k$, we seek to update $\boldsymbol{\alpha}_k$ according to its memberships $\pi_k$ as shown in Eq. (7). Hence, we have the following *code* update:

$$\boldsymbol{\alpha}_k^{\ell+1} = \boldsymbol{\alpha}_k^\ell - \pi_k (\boldsymbol{I} - \boldsymbol{Q}_k \boldsymbol{Q}_k^T) \boldsymbol{\alpha}_k^\ell \in \mathbb{R}^p \tag{12}$$

We then transform the *codes* into token feature by the following formula:

$$\boldsymbol{z}_{\ell+1} = \sum_{k=1}^K \boldsymbol{U}_k \boldsymbol{\alpha}_k^{\ell+1} \tag{13}$$

$$= \sum_{k=1}^K \boldsymbol{U}_k (\boldsymbol{\alpha}_k^\ell - \pi_k (\boldsymbol{I} - \boldsymbol{Q}_k \boldsymbol{Q}_k^T) \boldsymbol{\alpha}_k^\ell) \tag{14}$$

$$= \sum_{k=1}^K \boldsymbol{U}_k [\boldsymbol{U}_k^T \boldsymbol{z}_\ell - \pi_k (\boldsymbol{I} - \boldsymbol{Q}_k \boldsymbol{Q}_k^T) \boldsymbol{U}_k^T \boldsymbol{z}_\ell] \tag{15}$$

$$= \underbrace{\sum_{k=1}^K \boldsymbol{U}_k \boldsymbol{U}_k^T \boldsymbol{z}_\ell}_{\approx \boldsymbol{z}_\ell} - \sum_{k=1}^K \pi_k \boldsymbol{U}_k (\boldsymbol{I} - \boldsymbol{Q}_k \boldsymbol{Q}_k^T) \boldsymbol{U}_k^T \boldsymbol{z}_\ell \tag{16}$$

Based on the union of $K$ low-dimensional subspaces model mentioned above, where $\boldsymbol{z} = \boldsymbol{U}_k \boldsymbol{\alpha}$, Eq. (13) can be interpreted as follows: $\boldsymbol{U}_k \boldsymbol{\alpha}_k$ obtained in each subspace represents the component of the original token $\boldsymbol{z}$ in that subspace. Therefore, we can sum these components to recover the original $\boldsymbol{z}$. The approximation in Eq. (16) can be understood in the same manner.

## A.3 DETAILS ON OBTAINING THE BASIS OF COLUMN SPACE

To ensure the paper is self-contained, we briefly explain the steps of obtaining an orthogonal basis through Cholesky decomposition. For simplicity, we assume here that $\boldsymbol{Z}$ has already captured the column space by multiplication with the random matrix $\boldsymbol{\Omega}$. Given a Gram matrix $\boldsymbol{G} = \boldsymbol{Z}^T \boldsymbol{Z} \in \mathbb{R}^{m \times m}$, we can obtain the lower triangular matrix $L$ via Cholesky decomposition $\boldsymbol{G} = \boldsymbol{L} \boldsymbol{L}^T$. Hence, the lower triangular matrix $L$ can be used to calculate the orthogonal basis $\boldsymbol{Q}$ in the following way. Consider that $\boldsymbol{Q} \boldsymbol{Q}^T = \boldsymbol{I}$ and we seek to find a coefficient matrix $\boldsymbol{Q}_{\text{coef}}$ such that $\boldsymbol{Q} = \boldsymbol{Z} \boldsymbol{Q}_{\text{coef}}$. Then we have

$$(\boldsymbol{Z} \boldsymbol{Q}_{\text{coef}})^T (\boldsymbol{Z} \boldsymbol{Q}_{\text{coef}}) = \boldsymbol{I} \tag{17}$$

$$\boldsymbol{Q}_{\text{coef}}^T (\boldsymbol{Z}^T \boldsymbol{Z}) \boldsymbol{Q}_{\text{coef}} = \boldsymbol{I} \tag{18}$$

$$\boldsymbol{Q}_{\text{coef}}^T \boldsymbol{G} \boldsymbol{Q}_{\text{coef}} = \boldsymbol{I} \tag{19}$$

Since we have $\boldsymbol{G} = \boldsymbol{L}\boldsymbol{L}^T$, therefore

$$\boldsymbol{Q}_{\text{coef}}^T \boldsymbol{L}\boldsymbol{L}^T \boldsymbol{Q}_{\text{coef}} = \boldsymbol{I} \tag{20}$$

$$(\boldsymbol{L}^T \boldsymbol{Q}_{\text{coef}})^T (\boldsymbol{L}^T \boldsymbol{Q}_{\text{coef}}) = \boldsymbol{I} \tag{21}$$

Here, we can set $\boldsymbol{L}^T \boldsymbol{Q}_{\text{coef}} = \boldsymbol{I}$, thereby this upper triangular linear system of equations can be solved by backward substitution.

$$\boldsymbol{Q}_{\text{coef}} = (\boldsymbol{L}^T)^{-1} \tag{22}$$

$$\Rightarrow \boldsymbol{Q} = \boldsymbol{Z}\boldsymbol{L}^{-T} \tag{23}$$

### A.4 DETAILS ANALYSIS OF COMPLEXITY

Table 2: Computational steps and complexity analysis

| Step | Expansion Module | | | Compression Module | | |
|---|---|---|---|---|---|---|
| | **Matrix Operation** | **Time** | **Space** | **Matrix Operation** | **Time** | **Space** |
| 1 | $\boldsymbol{Y} = \boldsymbol{Z}\boldsymbol{\Omega}$ | $\mathcal{O}(dnKr)$ | $\mathcal{O}(nKr)$ | $\boldsymbol{Y}_k = \boldsymbol{A}_k\boldsymbol{\Omega}$ | $\mathcal{O}(pnr)$ | $\mathcal{O}(pn)$ |
| 2 | $\boldsymbol{M} = \boldsymbol{Y}^T\boldsymbol{Y}$ | $\mathcal{O}(dK^2r^2)$ | $\mathcal{O}(K^2r^2)$ | $\boldsymbol{M}_k = \boldsymbol{Y}^T\boldsymbol{Y}$ | $\mathcal{O}(pr^2)$ | $\mathcal{O}(r^2)$ |
| 3 | $\boldsymbol{L} = \text{ChoL}(\boldsymbol{M})$ | $\mathcal{O}(\frac{1}{3}K^3r^3)$ | $\mathcal{O}(K^2r^2)$ | $\boldsymbol{L}_k = \text{ChoL}(\boldsymbol{M}_k)$ | $\mathcal{O}(\frac{1}{3}r^3)$ | $\mathcal{O}(r^2)$ |
| 4 | $\boldsymbol{Q} = \boldsymbol{Y}\boldsymbol{L}^{-T}$ | $\mathcal{O}(dK^2r^2)$ | $\mathcal{O}(dKr)$ | $\boldsymbol{Q}_k = \boldsymbol{Y}_k\boldsymbol{L}_k^{-T}$ | $\mathcal{O}(pr)$ | $\mathcal{O}(pr)$ |
| 5 | $\boldsymbol{I} - \boldsymbol{Q}\boldsymbol{Q}^T$ | $\mathcal{O}(d^2Kr)$ | $\mathcal{O}(d^2)$ | $\boldsymbol{I} - \boldsymbol{Q}_k\boldsymbol{Q}_k^T$ | $\mathcal{O}(p^2r)$ | $\mathcal{O}(p^2)$ |
| **Total** | | $\mathcal{O}(dnKr)$ | $\mathcal{O}(nKr)$ | | $\mathcal{O}(pnr)$ | $\mathcal{O}(pn)$ |

## B EXPERIMENT DETAILS

### B.1 PRACTICAL IMPLEMENTATION

**Class attention layer for information aggregation.** For supervised classification task, a typical way to aggregate class-related information is by inserting a learnable [CLS] token in each layer. However, since ECAttention aims to compress different group features onto corresponding subspaces, inserting a [CLS] token at each layer may not be suitable as it do not belong to any subspaces, thereby cannot aggregate class-related information effectively. Hence, we use global class attention layer Ali et al. (2021) at the end of entire architecture.

**Numerical stability.** Since Cholesky decomposition can only be applied to positive definite matrices, we add a small identity matrix $\epsilon\boldsymbol{I}$ to the Gram matrix $\boldsymbol{Y}^T\boldsymbol{Y}$ to ensure numerical stability. In fact, after incorporating the identity matrix, the resulting projection matrix $\boldsymbol{Q}\boldsymbol{Q}^T$ is equivalent to obtaining a *regularized version* of the projection matrix. This is analogous to the operators in ReduNet, which have strong connections to ridge regression.

### B.2 MODEL CONFIGURATIONS

Table 3: The configurations and parameters of ECA

| | **ECA-T** | **ECA-S** | **ECA-M** | **ECA-B** | **ECA-B+** | **ECA-L** |
|---|---|---|---|---|---|---|
| # parameters | 1.91M | 7.2M | 14.7M | 21.0M | 57.7M | 70M |
| # attention heads $K$ | 4 | 8 | 8 | 16 | 16 | 16 |
| # layers $L$ | 12 | 12 | 24 | 48 | 24 | 36 |
| # feature dimension $d$ | 192 | 384 | 512 | 512 | 1024 | 1024 |
| # head dimension $p$ | 48 | 48 | 64 | 64 | 64 | 64 |

Table 4: The configurations and parameters of ToST

|  | ToST-T | ToST-S | ToST-M |
|---|---|---|---|
| # parameters | 5.8M | 22.6M | 68.1M |
| # attention heads $K$ | 4 | 8 | 8 |
| # layers $L$ | 12 | 12 | 24 |
| # feature dimension $d$ | 192 | 384 | 512 |
| # head dimension $p$ | 48 | 48 | 64 |

Table 5: The configurations and parameters of DeiT

|  | DeiT-T | DeiT-S | DeiT-B |
|---|---|---|---|
| # parameters | 5M | 22M | 86M |
| # attention heads $K$ | 3 | 6 | 12 |
| # layers $L$ | 12 | 12 | 12 |
| # feature dimension $d$ | 192 | 384 | 768 |
| # head dimension $p$ | 64 | 64 | 64 |

Table 6: The configurations and parameters of CRATE

|  | CRATE-Tiny | CRATE-Small | CRATE-Base | CRATE-Large |
|---|---|---|---|---|
| # parameters | 6.1M | 13.1M | 22.8M | 77.6M |
| # attention heads $K$ | 6 | 12 | 12 | 16 |
| # layers $L$ | 12 | 12 | 12 | 24 |
| # feature dimension $d$ | 384 | 576 | 768 | 1024 |
| # head dimension | 32 | 48 | 64 | 64 |

Due to the quadratic time complexity of CRATE and DeiT, as well as limited computational resources, our experiments primarily compare against ToST. For CRATE and DeiT models, we directly use the results reported in their respective papers(Yu et al., 2023; Touvron et al., 2021). As shown in Table 3, we tested ECA models with six different parameters scales including ECA-T(Tiny), ECA-S(Small), ECA-M(Medium), ECA-B(Base), ECA-B+(Base+) and ECA-L(Large). Note that under identical settings, our model actually has significantly fewer parameters than ToST as shown in Table 3 and Table 4.

## B.3 THE RESULTS OF DEIT

Table 7: Top 1 accuracy of DeiT across datasets with different model sizes.

| Datasets | DeiT-T | DeiT-S | DeiT-B |
|---|---|---|---|
| # parameters | 5M | 22M | 86M |
| ImageNet | 72.2 | 79.8 | 81.8 |
| ImageNet ReaL | 80.1 | 85.7 | 86.7 |
| CIFAR10 | - | - | 99.1 |
| CIFAR100 | - | - | 90.8 |
| Oxford Flowers-102 | - | - | 98.4 |
| Oxford-IIIT-Pets | - | - | - |

## B.4 TRAINING SETUP

Table 8: Batch size settings for different ECA model scales

| Model | Batch Size |
|-------|-----------|
| ECA-T | 2048 |
| ECA-S | 1024 |
| ECA-M | 512 |
| ECA-B | 256 |
| ECA-B+ | 256 |
| ECA-L | 128 |

**Pre-training on ImageNet-1k.** Our models are trained using the AdamW optimizer with a learning rate of $5 \times 10^{-4}$ for $500$ epochs. Due to computational resource constraints, we used different batch sizes for models of different scales as shown in Table 8. All images are resized to $224 \times 224$ resolution and divided into $16 \times 16$ patches, which are then converted into patch embeddings. For other training configurations, we follow the same protocol as in (Wu et al., 2025). All pre-training experiments are conducted on 2 NVIDIA H100 GPUS.

**Fine-tuning.** Fine-tuning experiments are conducted using pretrained ECA and ToST models as initialization, then fine-tuning them on the following datasets: CIFAR-10/100 (Krizhevsky et al., 2009), Oxford Flowers (Nilsback & Zisserman, 2008), and Oxford-IIIT Pets (Parkhi et al., 2012). All fine-tuning experiments employ a batch size of 256 with a learning rate of $1 \times 10^{-4}$.

## C ADDITIONAL EXPERIMENTAL STUDIES

### C.1 REGULATE EXPANSION–COMPRESSION DYNAMICS VIA ENFORCING WEIGHT POSITIVITY

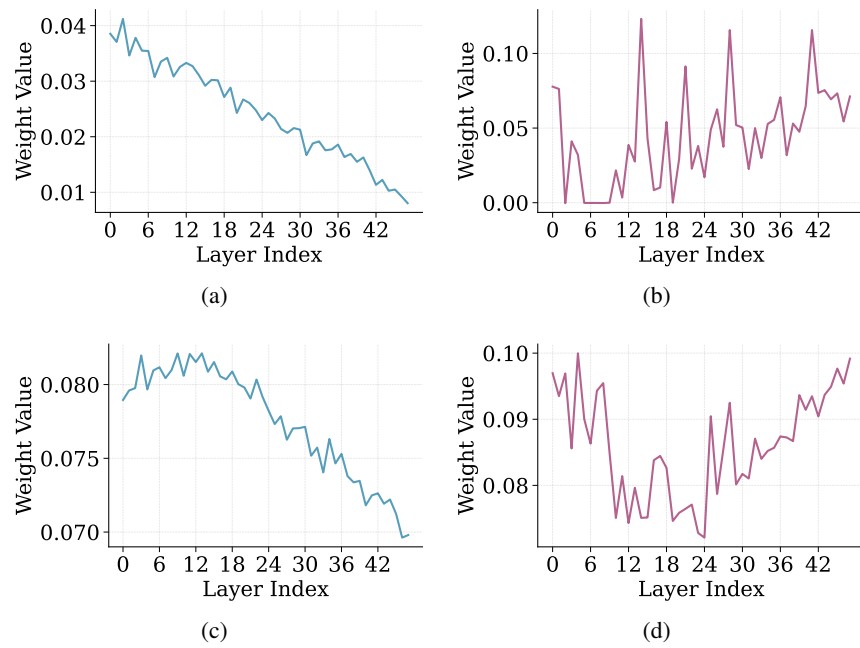

Figure 11: The weight distribution of ECA-B after training on CIFAR-10. (a) Weight of the expansion module. (b) Weight of the compression module. (c) Weight of the expansion module after enforcing weight positivity. (d) Weight of the compression module after enforcing weight positivity.

As shown in Figure 11, after applying the softplus function (Dugas et al., 2000) to enforce weight positivity, the relative fluctuations of the weights across layers are reduced, as evidenced by the narrower vertical axes in Figures 11(c) and 11(d). Moreover, all layers are activated and contribute to feature updates, unlike in Figure 11(b), where some layers are effectively inactive.

## C.2 A FAIRER COMPARISON WITH ToST ON CIFAR-100

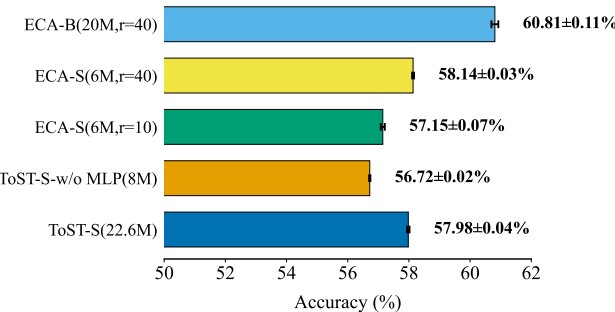

Figure 12: A fairer comparison with ToST on CIFAR-100 after 100 epochs training.

As shown in Figure 12, we also conduct a comparison with ToST on the CIFAR-100 dataset. This experiment further confirms that ToST's superior performance mainly stems from its MLP module. Notably, our ECA-S (6M) model surpasses ToST-S (22.6M) with a hyperparameter $r = 40$. Due to the time constraints of the rebuttal and limited computational resources, we are unable to conduct larger-scale experiments in the short term. Nevertheless, we believe that the current experiments are sufficient to demonstrate the promise of our model.

## C.3 ABLATION STUDY ON EXPANSION MODULE

We evaluate the impact of the expansion module using ECA-S on CIFAR-10 over 400 training epochs. As shown in Table 9, the model with the expansion module exhibits better accuracy.

Table 9: Accuracy (%) with and without the expansion module.

| Dataset | w/o Expansion | w/ Expansion |
|---|---|---|
| CIFAR-10 | 90.40 | **90.71** |

## C.4 COMPRESSION VIA DIRECT COLUMN SPACE PROJECTION VS. NULL SPACE SUBTRACTION

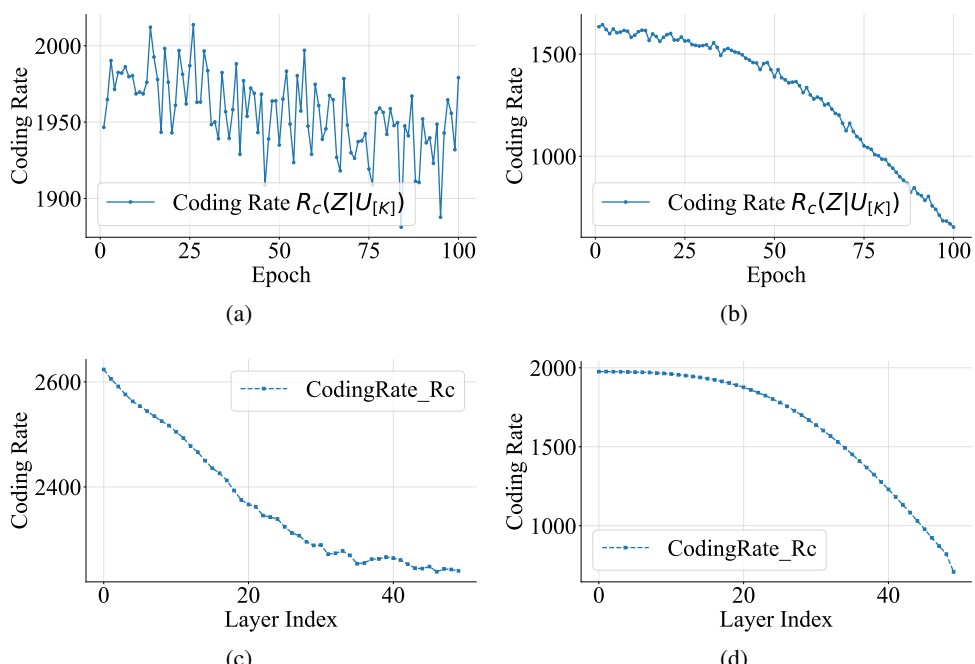

Figure 13: (a) Compression via directly projecting onto the column space. (b) Compression via subtracting the null space projection. (c) Layer-wise compression via directly projecting onto the column space. (d) Layer-wise compression via subtracting the null space projection.

We conduct this experiment on toy data, as shown in Figure 13(b), compression via subtracting the null space projection enables smoother optimization. In contrast, directly projecting onto the column space is not very effective, either in terms of layer-wise compression or multi-epoch optimization.

## C.5 SENSITIVITY EXPERIMENTS ON REGULARIZATION TERM IN CHOLESKY DECOMPOSITION.

Table 10: Ablation study on the regularization term $\epsilon$. We report the results after 40 training epochs on the CIFAR-10 dataset.

| $\epsilon$ | Time/Epoch (m:s) | Fail Rate (%) | Acc (%) |
|---|---|---|---|
| 0 | - | - | - |
| $10^{-4}$ | 8:12 | 21.28 | 50.11 |
| $10^{-3}$ | 7:33 | 11.11 | 71.96 |
| $10^{-2}$ | 0:55 | 0 | 71.80 |
| $10^{-1}$ | 0:55 | 0 | 71.59 |

A common trick to ensure numerical stability during Cholesky decomposition is to add a small identity matrix term $\epsilon \mathbf{I}$ to the diagonal. As shown in Table 10, we evaluate the failure rate of Cholesky decomposition (i.e., when it falls back to the more expensive QR decomposition), the corresponding Top-1 accuracy, and the per-epoch training time under different values of $\epsilon$. When $\epsilon = 10^{-2}$, Cholesky decomposition never fails, while delivering both reasonable training speed and accuracy. This is therefore adopted as the default setting in our model. In contrast, setting $\epsilon = 0$ (i.e., no stabilization) quickly leads to NaN values during training, making the model untrainable.

For smaller values such as $\epsilon = 10^{-3}$ and $\epsilon = 10^{-4}$, we observe that the failure probability of Cholesky decomposition significantly increases. Consequently, the training process frequently falls back to QR decomposition, resulting in a substantial increase in per-epoch training time.

### C.6 MORE ATTENTION VISUALIZATIONS

Here, we show visualizations of attention from more layers. Due to layer-wise compression, deeper layers may exhibit fragmented attention. Nevertheless, different heads in each layer still focus on distinct parts of the image.

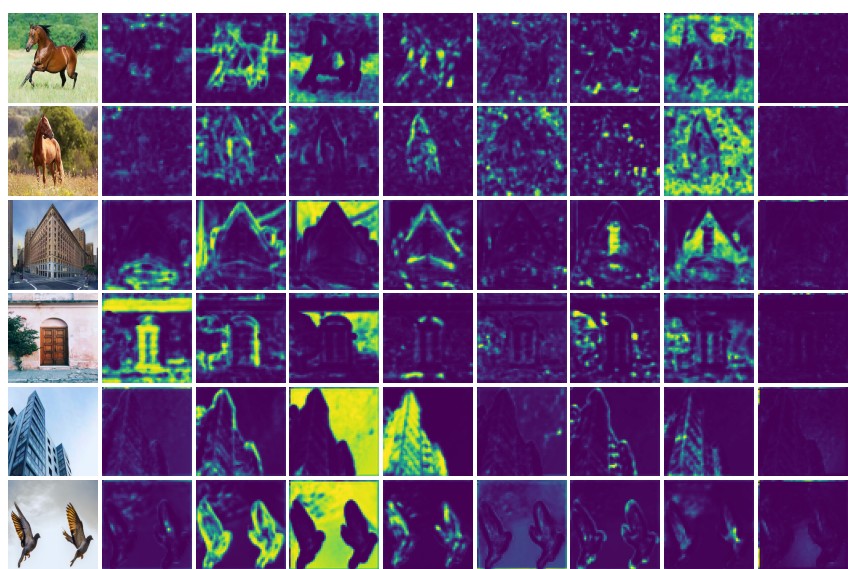

Figure 14: **Membership distribution $\pi$ of ECA-B. Results are shown for layer 2.**

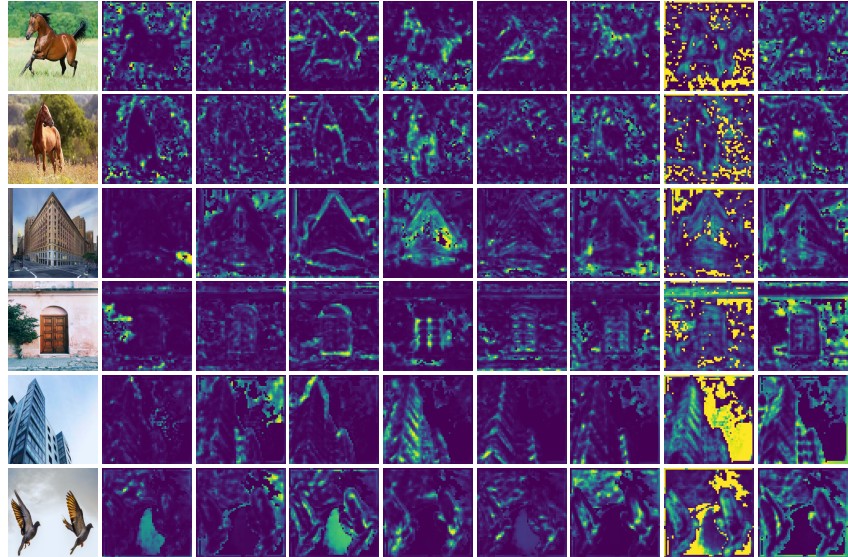

Figure 15: **Membership distribution $\pi$ of ECA-B. Results are shown for layer 20.**

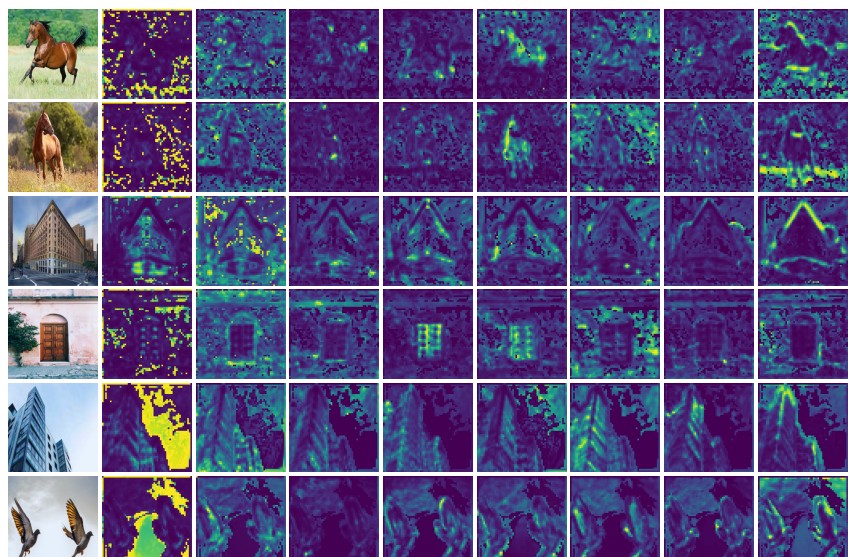

Figure 16: **Membership distribution $\pi$ of ECA-B. Results are shown for layer 40.**

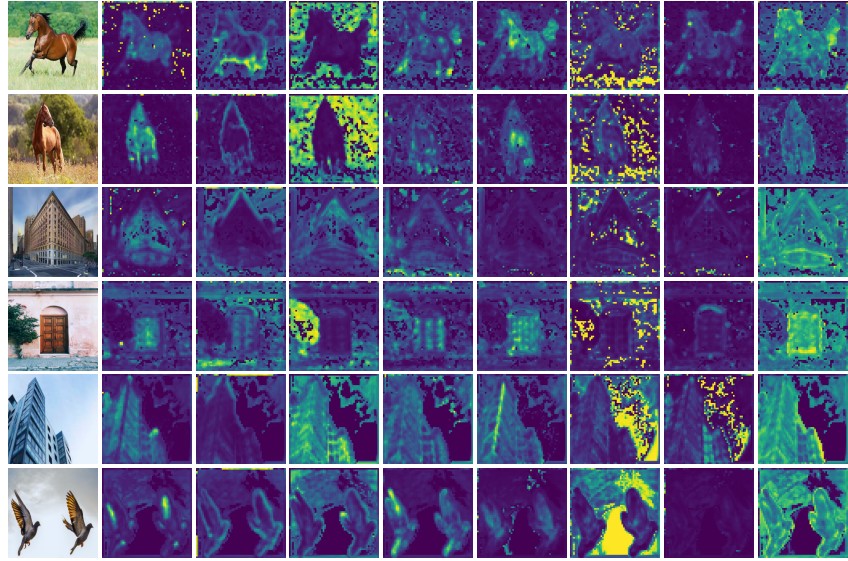

Figure 17: **Membership distribution $\pi$ of ECA-B. Results are shown for layer 48.**

## D  PSEUDOCODE OF ECATTENTION

See Algorithms 1, 2, 3 and 4 on the following pages.

## E  LIMITATIONS

Despite the promising results, this work has several limitations. Although the framework shows some adaptability to random matrix size, this study does not quantify the potential information loss under varying matrix sizes. Besides, due to limitations in computational resources and time, it remains to be investigated whether our framework can achieve strong performance on larger-scale tasks.

**Algorithm 1** Expansion module of ECAttention in PyTorch

```python
class Expand(nn.Module):
    def __init__(self, dim, heads, subspace_rank):
        super().__init__()
        self.dim = dim
        self.heads = heads
        self.rank = subspace_rank * heads
        # Random matrix for capturing column space
        self.register_buffer(
            'Omega', torch.randn(n_tokens, self.rank)
        )

    def forward(self, x):
        b, n, d = x.shape
        x_t = x.transpose(-1, -2)

        # capturing column space
        Y = torch.matmul(x_t, self.Omega)

        # Obtaining the basis of Y via Cholesky decomposition
        Q = cholesky_orthogonalization(Y)

        # Null space projection
        QQT = torch.matmul(Q, Q.transpose(-1, -2))
        null_proj = x_t - torch.matmul(QQT, x_t)

        return null_proj.transpose(-1, -2)
```

## F   USE OF LARGE LANGUAGE MODELS

We declare that large language models (LLMs) were used solely for language polishing in this paper. All ideas, methods, experiments, and conclusions are the authors' own, and no substantive content was generated by LLMs.

**Algorithm 2** Compression module of ECAttention in PyTorch

```python
class Compress(nn.Module):
    def __init__(self, dim, heads, dim_head, subspace_rank):
        super().__init__()
        self.heads = heads
        self.subspace_rank = subspace_rank
        self.Us = nn.Parameter(torch.randn(dim, dim_head * heads))
        self.temperature = nn.Parameter(torch.ones(1))
        # Random matrix for capturing column space
        self.register_buffer(
            'Omega', torch.randn(n_tokens, subspace_rank)
        )

    def forward(self, x):
        b, n, d = x.shape
        x_t = x.transpose(-1, -2)

        Us_T = rearrange(self.Us, 'd (h p) -> h p d', h=self.heads)
        alphas = torch.einsum('hpd,bdn->bhpn', Us_T, x_t)

        # Per-head null space projection
        Qs = []
        for i in range(self.heads):
            Y_i = torch.einsum(
                'bpn,nc->bpc', alphas[:, i], self.Omega
            )
            Q_i = cholesky_orthogonalization(Y_i)
            Qs.append(Q_i)

        Qs = torch.stack(Qs, dim=1)
        QQT = torch.matmul(Qs, Qs.transpose(-1, -2))

        # Column space and null space projections
        col_proj = torch.einsum('bhpp,bhpn->bhpn', QQT, alphas)
        null_proj = alphas - col_proj

        # Attention-based weighting
        norms = torch.linalg.norm(col_proj, dim=-2)
        weights = F.softmax(norms / self.temperature, dim=-1)

        # Weighted aggregation and reconstruction
        w_null_proj = torch.einsum(
            'bnh,bhpn->bhpn', weights, null_proj
        )
        w_null_proj_re = rearrange(
            w_null_proj, 'b h p n -> b (h p) n'
        )
        output = torch.matmul(self.Us, w_null_proj_re)

        return output.transpose(-1, -2)
```

**Algorithm 3** ECAttention in PyTorch

```python
class ECAttention(nn.Module):
    def __init__(self, dim, num_heads, subspace_rank):
        super().__init__()
        self.expand = Expand(dim, num_heads, subspace_rank)
        self.compress = Compress(
            dim, num_heads, dim//num_heads, subspace_rank
        )

        self.gamma1_raw = Parameter(
            log(exp(tensor(eta1)) - 1.0)
        )
        self.gamma2_raw = Parameter(
            log(exp(tensor(eta2)) - 1.0)
        )

    # Expand coefficient
    @property
    def alpha(self):
        return softplus(self.gamma1_raw)

    # Compress coefficient
    @property
    def beta(self):
        return softplus(self.gamma2_raw)

    def forward(self, x):
        # x: (batch, n_tokens, dim)
        expand_out = self.expand(x)
        compress_out = self.compress(x)

        # Residual connection with learnable coefficients
        x = x + self.alpha * expand_out - self.beta * compress_out
        return x
```

**Algorithm 4** Cholesky decomposition step in PyTorch

```python
def cholesky_orthogonalization(Y, eps=0.01):
    """
    Obtaining the basis of the column space using Cholesky
    decomposition.

    Args:
        Y: Input matrix of shape (batch, dim, rank)
        eps: Regularization parameter for numerical stability

    Returns:
        Q: Orthogonalized matrix of shape (batch, dim, rank)
    """
    # Normalize columns for numerical stability
    Y_norm = F.normalize(Y, p=2, dim=-2)

    try:
        # Compute Gram matrix: G = Y_norm^T @ Y_norm
        G = torch.matmul(Y_norm.transpose(-1, -2), Y_norm)

        # Add regularization: G = G + eps * I
        I = torch.eye(G.size(-1), device=G.device)
        G_reg = G + eps * I

        # Cholesky decomposition: G_reg = L @ L^T
        L = torch.linalg.cholesky(G_reg)

        # Solve triangular system: L^T @ C = I
        C = torch.linalg.solve_triangular(
            L.transpose(-1, -2), I, upper=True
        )

        # Compute orthogonal matrix: Q = Y_norm @ C
        Q = torch.matmul(Y_norm, C)

    except RuntimeError:
        # Fallback to QR decomposition if Cholesky fails
        Q, _ = torch.linalg.qr(Y_norm, mode="reduced")

    return Q
```

