# OpenReview forum: "ECAttention: Adaptive and Principled Feature Expansion-Compression with Linear Efficiency"
_ICLR.cc/2026/Conference — Submitted to ICLR 2026_

### Official Review · Reviewer_vfW8 · 2025-10-28

**Soundness:** 2
**Presentation:** 2
**Contribution:** 2
**Rating:** 4
**Confidence:** 4

**Summary:**

This paper proposes ECAttention, a linear-time attention mechanism inspired by the Maximal Coding Rate Reduction ($MCR^2$) objective. The authors argue that previous $MCR^2$-based models, like CRATE and ToST, are either unprincipled in their approximation or over-simplify the problem. The core idea of ECAttention is to use a geometric insight—projecting features onto their column space and null space—to perform expansion and compression. The method uses randomization and Cholesky decomposition to efficiently find the basis of these subspaces, achieving linear-time complexity relative to the number of tokens ($n$). The model also introduces two learnable parameters ($\alpha, \beta$) per layer to "self-adaptively" balance the strength of expansion and compression. The authors claim this results in a more principled and interpretable model that achieves comparable or-superior performance to baselines like ToST and CRATE.

**Strengths:**

1. The paper's strongest point is Figure 5. The membership visualizations for ECA-B are qualitatively excellent, demonstrating a fine-grained, semantically meaningful segmentation of objects (e.g., separating a building's floors, or a tree from a door). This provides strong evidence for the authors' claim that ToST's simplification (ignoring inter-dimensional correlations) misses important structural information, and that ECAttention's approach successfully captures it.

2. The paper plots inference time and peak memory vs. tokens and provides an appendix complexity breakdown; CRATE is quadratic, while ECA trends linearly, with an added Cholesky cost.

3. ECA-S (7.2M params) achieves 79.0% on ImageNet-ReaL, beating CRATE-L; other small/medium comparisons to ToST are “comparable.”

**Weaknesses:**

1. Does not clearly surpass the strongest baselines at scale. The paper itself concedes ECA “falls slightly behind ToST at larger parameter scales,” partly blaming a fixed rank (r=20). For a most-competitive venue, “comparable” is usually insufficient—particularly when the method adds nontrivial machinery (random projections + Cholesky) on top of ToST-like scaffolding. Strong, fair, large-scale sweeps (varying (r), compute-matched) are needed.

2. Linear-time claim depends on a non-negligible Cholesky subroutine. The paper frames the overall complexity as linear in tokens with a “minor fixed overhead,” yet implements an r×r Cholesky per head/layer (plus fallbacks), which can easily become the hidden bottleneck depending on r, heads, and batch size. The current analysis underplays these constants and their training-time impact.

3. The Cholesky path needs regularization and has a QR fallback on failure; the paper doesn’t quantify how often Cholesky fails, how ε and temperature affect training.

4. The authors note the “expansion” weight can go negative (thus compressing) when r is large; this blurs the conceptual neatness of separate expansion vs. compression and suggests under-constrained dynamics. Conditions for when expansion is beneficial vs. harmful are not established.

5. There is no evaluation on long-sequence tasks that would stress the purported advantages of linear scaling.

**Questions:**

The idea is clean and well-motivated, and the qualitative behavior is appealing. However, the empirical section does not yet establish a decisive advantage over the strongest baselines at relevant scales, and important practical/theoretical questions remain open (stability, overheads, and regime-dependent behavior). With stronger large-scale results, robustness analyses, this could move to an accept and I am willing to reevaluate accordingly.

---

> ### Author Response · Authors · 2025-11-21
>
> Thanks for your careful reading and valuable comments.
> We focus here on the issues raised regarding our experiments and provide detailed responses and corresponding revisions.
>
> **Does not clearly surpass ToST?**
>
> We observed that **the strong performance of ToST reported in Table 1 is largely attributable to its non-white-box MLP module.**
> This is also reflected in Table 8 of the ToST paper’s appendix [1]. We provide additional explanation in Section 4.2, and in Section 4.4, we include a fairer comparison with ToST.
> Specifically, the MLP module accounts for nearly two-thirds of ToST’s parameters. In Figure 9, we show that removing the MLP module reduces ToST-S (22M) to 8M parameters (denoted as ToST-S-w/o MLP).
> Our proposed model, ECA-S (6M) with $r=10$ (accuracy on CIFAR-10 is 81.61\%), already surpasses the performance of ToST-S-w/o MLP (81.16\%).
> Moreover, **increasing $r$ further improves accuracy**, as evidenced by ECA-S with $r=40$, which achieves 82.14\% accuracy.
> As shown in Figure 8b, by applying the softplus trick[2] to ensure positive weights for the compression and expansion modules, the trend of higher accuracy with larger $r$ becomes more stable.
>
> In summary, compared with ToST, the superiority of our method lies in that the compression modules at each layer can capture correlation among dimensions for feature updates, achieving better interpretability (as evidenced by Figure 6) and higher classification accuracy (as evidenced by Figure 9), while incurring only a marginal increase in computational complexity compared to ToST.
>
> Due to time and computational resource limitations, we conducted the experiments with different $r$ values on CIFAR-10 and CIFAR-100 (Appendix C.2). Performing these experiments on large-scale datasets such as ImageNet is infeasible with our current resources within a short timeframe. Nevertheless, we believe that the existing experimental results are sufficient to demonstrate that our method further advances this line of research and makes a meaningful contribution.
>
> **The complexity analysis of Cholesky**
>
> Since the compression and expansion modules represent the minimal complete units
> of the network, the complexity analysis provided in Appendix A.4, Table 2,
> is sufficient. The complexities of CRATE, ToST, and even ViT can also be
> analyzed in a similar manner, focusing on the attention module as the minimal
> complete unit without including the batch size.
>
> As shown in Figure 2, our model achieves near-linear complexity compared to
> the baselines. Including the batch size would only scale the complexity of
> all models by a constant factor.
>
> Regarding the number of heads $K$, as analyzed in Table 2, the time
> complexity of the expansion module is $O(dnKr)$, and that of the
> compression module is $O(pnr)$. Since $K \ll n$, the overall complexity
> of the model is primarily dominated by the number of tokens $n$ and the
> feature dimension $d$ or subspace dimension $p$.

---

> ### Author Response · Authors · 2025-11-21
>
> **Failure rate of Cholesky**
>
> We set $\epsilon = 10^{-2}$ as the default value for the regularization term in the Cholesky decomposition. Under this setting, the Cholesky decomposition exhibits a failure rate of 0.
>
>
> In Appendix C.5, we present
> a study on the fallback of Cholesky decomposition to QR when it fails.
> Specifically, when the regularization term $\epsilon$ in Cholesky decomposition
> is set to $10^{-3}$, the failure rate is 11.11\%, and the training time per
> epoch on CIFAR-10 is 7 minutes and 33 seconds due to fallback to QR. In contrast,
> setting $\epsilon = 10^{-2}$ results in a 0\% failure rate, with each epoch
> taking less than one minute.
> As for the temperature, it is a learnable parameter in our model; therefore,
> we did not conduct an ablation study on it.
>
> **Ablation study on the regularization term ε.**
> We report the results after 40 training epochs on the CIFAR-10 dataset.
>
> | ε        | Time/Epoch (m:s) | Fail Rate (%) | Acc (%) |
> |----------|-----------------|---------------|---------|
> | 0        | -               | -             | -       |
> | 1e-4     | 8:12            | 21.28         | 50.11   |
> | 1e-3     | 7:33            | 11.11         | 71.96   |
> | 1e-2     | 0:55            | 0             | 71.80   |
> | 1e-1     | 0:55            | 0             | 71.59   |
>
> **Constrain dynamics of expansion and compression.**
> This is a heuristic design choice, intended to give the model more flexibility rather than strictly constraining its dynamics.
> Directly optimizing the weights of the expansion module may result in negative values, which could prevent the module from performing actual expansion. To better control the dynamics---ensuring that the expansion module truly expands and the compression module truly compresses---we apply the **softplus function**[2] to constrain the weights to be positive. This is a common technique widely used in deep learning.
>
> We present the related experiments in Figure 8 and Appendix C.1. Before applying the softplus function, we observed that when $r$ was relatively small, some compression layers were not activated, with their weights being zero. After using softplus, all layers were guaranteed to be active. Moreover, we found that applying softplus reduced the fluctuations of the weights in both compression and expansion modules across layers, as evidenced by Figure 8(a). Additionally, the trend that higher $r$ leads to higher accuracy became more stable and controllable after introducing softplus, as evidenced by Figure 8(b).
>
> **long-sequence tasks?**
> We appreciate the suggestion and hope to explore the application of our method to long-sequence tasks in natural language in future work. Currently, we focus on image data to validate the theoretical design of our approach.
>
> **References:**
>
> [1] Wu, Ziyang, et al. "Token Statistics Transformer: Linear-Time Attention via Variational Rate Reduction." The Thirteenth International Conference on Learning Representations.
>
> [2] Dugas, Charles, et al. "Incorporating second-order functional knowledge for better option pricing." Advances in neural information processing systems 13 (2000).

---

> > ### Comment · Reviewer_vfW8 · 2025-11-26
> >
> > I thank the authors for their detailed response and the revisions made to the manuscript.
> >
> > The authors have effectively addressed my primary concerns regarding the technical stability and comparative fairness of the proposed method.
> >
> > While these issues have been resolved, I will withhold finalizing my score for the moment. I intend to see the comments and ongoing discussions from the other reviewers to ensure there are no remaining critical flaws I may have overlooked before making my final recommendation.

---

> > > ### Author Response · Authors · 2025-11-27
> > >
> > > We sincerely thank Reviewer  vfW8 for recognizing our responses and for the helpful comments.

---

### Official Review · Reviewer_6NA9 · 2025-10-29

**Soundness:** 1
**Presentation:** 1
**Contribution:** 2
**Rating:** 2
**Confidence:** 4

**Summary:**

Authors propose a deep (residual) neural network that, at every layer, employs two modules: an expansion module (that encourages the layer to project the latent variables onto a higher rank); and a compression module (that brings closer the representation of latent vectors that are already similar). Authors evaluate the method empirically (classification on CIFAR and Imagenet) and also qualitatively (by visualizing attention map per attention head).

Overall, the paper is potentially a very good contribution. But the presentation is not ripe yet. The authors may reconsider restructuring the paper and re-submitting at a future time. I will not recommend the paper to be accepted as-is, nor will I review a completely different version. I will unlikely change my opinion with just simple write-up improvements, as the paper (and experiments) deserve a complete re-write for consideration.

**Strengths:**

* There can be several motivations of this work, including:
  * Whitebox models: visualizing saliency maps can inform users what part of the inputs is the model latching on.
  * Unsupervised learning can create some latent space that can be used later for supervised tasks.
  * Potentially useful for training compression models.
* The results are competitive, yet the training is fast.

**Weaknesses:**

# Notation is confusing
* Why not clearly put an equation of what $\mathbf{Q}$ equals to? The text above equation 5 says it is an orthonormal basis for $\mathbf{Z}$. In that case, shouldn't be a function of $\ell$ also?
* Line 119 defines $U_{[K]}$... What is $\in \mathbb{R}^{d \times K_p}$? Is it the entire $U_{[K]}$ or the constituents $U_s$? The notation sas the first but the intuition says the second. **please** be more exact.
* The paper mentions what is $\mathbf{Q}$ (orthonormal basis for $\mathbf{Z}$) but never mentions what is $\mathbf{Q}_k$. I spent a fair amount of time to *guess* what is $\mathbf{Q}_k$ and I think I have a great guess! It is exactly the orthonormal basis of $U_k \mathbf{Z}$. Am I right? I deserve a brownie! But why not explicitly mention it? Trade handwavey informal text for method completeness and conciseness.
* Line 210: What is "each feature" $z \in \mathbf{Z}$? Is $\mathbf{Z}$ a matrix  and you are iterating over every row $z$ of $\mathbf{Z}$?
* The paper abruptly shifts from matrix form $\mathbf{Z}$ to vector form $z$. Why not stick to matrix form? This way, your $\pi_k$'s are always vectors rather than suddenly becoming scalars, etc.

# Unsupported Claims
* "**principled**" the paper keeps mentioning that other methods are not principled but the proposed method is principled. Please be specific. Principled in what way? What properties must be satisfied by a "principled method"? The paper must present proofs to show that the proposed method is principled, per whatever way the paper chooses to define "principled", and then show why other methods do not satisfy the definition.
* "**nearly linear complexity**" mentioned in abstract is not supported. Isn't it exactly linear, if one assumes that $K << n$? Either way, you should have some note about the complexity in the main paper, even if the derivation/proof is delayed until the appendix.


# Writing is unfit for ICLR
* You use word "thus" on line 150. IU would expect some proof or reference. Otherwise, downplay and use "could" or "should".
* Equation 3 has a math bug. Last row should use subscript $K$.
* The paper uses too much handwavy text and not much math. Notation is an art: it is supposed to deliver the information without using too many words. Consider doing so especiallly in section 3.
* Writing is too low-level for first paragraph of the intro. Why are we diving so deep in related work already?
* The application domain -- "images" is only mentioned at the last sentence of the abstract, and not mentioned in intro. It makes me want to believe that the method is "more general than just images" but there are no experiments to support that.
* The word "respectively" on line 72 seems inappropriately used.
* Line 102: "stands for the probability"... please be more specific. The probability is a scalar and not a vector.

# Experiments are not compelling
* There is no experiment that pushes me to want to use this method immediately. Perhaps it is under-sold or creativy is needed for better sales pitch? When and why would I use this method?
* Figure 5 visualizes on layer of each model. Does it make sense to visualize more layers?
* Table 2 does not mention the dataset it was conducted over.

**Questions:**

* Why have Cholesky Decomposition in the Abstract? It seems that it is only to find some matrix decomposition as a subroutine for Randomized SVD?

* Is it that for every $i \in [n]$ we have $\sum_{j \in [K]} \Pi_{i j} = 1$? If so, why not explicitly add that ear Equation 1?

* Is it possible to show image reconstructions?

* Why is "regularization" needed for cholesky decomposition? It definitely changes the charecteristics of the gram matrix. I assumed the normalization (line 994) is sufficient to make Y^T Y full-rank i.e. succeeding the "Cholesky" and the "solve_triangle" instructions.

---

> ### Author Response · Authors · 2025-11-21
>
> Thank you for recognizing my contribution and for your helpful suggestions on the presentation. I truly appreciate your taste and attention to detail—especially regarding mathematical notation—and I have learned a great deal from your feedback. Based on your comments, we have made several revisions. Here, we first
> address the questions you raised, followed by our responses to the weaknesses
> you pointed out.
>
> ## Questions
> **Why have Cholesky Decomposition in the Abstract?**
> The Cholesky decomposition is crucial for obtaining the feature column space
> at low cost. Using alternative matrix factorizations, such as QR decomposition,
> would significantly increase computational cost. In Appendix C.5, we present
> a study on the fallback of Cholesky decomposition to QR when it fails.
> Specifically, when the regularization term $\epsilon$ in Cholesky decomposition
> is set to $10^{-3}$, the failure rate is 11.11\%, and the training time per
> epoch on CIFAR-10 is 7 minutes and 33 seconds due to fallback to QR. In contrast,
> setting $\epsilon = 10^{-2}$ results in a 0\% failure rate, with each epoch
> taking less than one minute.
>
> **$\sum_{j\in[K]}{\Pi}_{ij} = 1$?** Yes, we have made the corresponding modifications in the main text.
>
> **Is it possible to show image reconstructions?**
> Yes, essentially, the layer-wise compression can be regarded as a denoising process. In the CRATE paper, a connection between denoising and diffusion processes was established. In future work, we may further explore applications in image reconstruction.
>
> **Why is "regularization" needed for cholesky decomposition?**
> Thank you for reading our paper so carefully, and even examining our code.
> Theoretically, $Y^\top Y$ only guarantees a positive semi-definite matrix,
> whereas Cholesky decomposition requires a positive definite matrix.
> Therefore, we add
> a regularization term to ensure that the decomposition proceeds correctly.
> In Appendix C.5, we provide experimental validation of the necessity of this
> regularization term. When $\epsilon = 0$, training is impossible. In contrast,
> setting $\epsilon = 0.02$ results in a 0\% failure rate for Cholesky decomposition.
>
> ## Notation
> We define $Z = [z_1, \dots, z_n] \in \mathbb{R}^{d \times n}$ in Section 2, where each column vector represents a token. For clarity and brevity of presentation, we suppress the layer index $\ell$ on all layer-specific parameter matrices (such as $Q_{(\ell)}$, $U_{(\ell)}$, etc.).
> For the same reason---namely, to keep the main text focused on the core ideas—we did not explicitly define $Q$ in the main body. The complete definition is provided in Appendix A.3 (Eq. 23), where
> $Q = Z L^{-T}$, and $L$ is  result of the Cholesky decomposition of the Gram matrix $Z^\top Z$.
> And yes, your understanding regarding on $Q_k$ is correct. (If we ever meet in person, I will bring you a brownie!).
> In the earlier version, I mistakenly postponed the explanation of $Q_k$ to Section 3.2. This has now been corrected, and we include the clarification immediately after Eq. 4 in Section 3.1.
>
> In addition, $U_{[K]} \in \mathbb{R}^{d \times Kp}$ refers to the concatenation of all $K$ subspace bases. Each $U_i \in \mathbb{R}^{d \times p}$ (with $p < d$) spans the basis of the $i$-th low-dimensional subspace. We have added this explanation above Eq. 2 in the revised manuscript.
>
> Finally, we use the vector form $z$ rather than the matrix form $Z$ mainly to make the update process of each individual feature explicitly clear. Representing the updates in a matrix form would obscure the step-by-step behavior of our attention mechanism and make it more difficult to see how the updates are actually applied.

---

> > ### Author Response · Authors · 2025-11-21
> >
> > ## Claims on principled and nearly linear complexity
> >  **Definition of *principled*.**
> > In this paper, we use the term *principled* to indicate that the
> > network architecture and its layer-wise update rules can be
> > derived from a well-defined optimization objective. Specifically, our
> > method is grounded in the expansion objective $R(Z)$ in Eq. (1) and the compression objective in Eq. (2). Under this definition,
> > a principled feature update should follow the gradient (or a faithful
> > approximation of the gradient) of these objectives, ensuring that the
> > update direction is theoretically justified rather than heuristic.
> > In our paper, we construct the expansion and compression modules at each
> > layer based on the geometric interpretation of the gradient of the
> > corresponding objectives. This design enables layer-wise feature expansion
> > or compression, as illustrated in Figure 3.
> >
> > **Why other methods do not satisfy the definition.**
> > We discuss the limitations of MSSA and TSSA below Eq. 3 in Section 2 as part
> > of the motivation for our work. In brief, the MSSA module in CRATE is a
> > second-order approximation of the gradient of the compression objective
> > (Eq. 2), but it discards the first-order term, which causes its update
> > behavior to violate the intended compression principle.
> >
> > The TSSA module in ToST is overly simplified: its proposed upper-bound
> > surrogate of Eq. 2 uses only the diagonal elements of the feature correlation
> > matrix to guide updates, ignoring inter-dimensional correlations among the data.
> >
> > Furthermore, as we show in Section 4.4, the superior performance of ToST is
> > mainly due to the MLP module following TSSA, which accounts for nearly
> > two-thirds of the total parameters. These observations reinforce the need for
> > a principled, gradient-based design as in our approach.
> >
> > **Nearly linear complexity**
> > Before Section 4, we provide both a complexity analysis and visualization.
> > The computational complexity mainly depends on the number of tokens $n$. As
> > shown in Figure 2a, when $n$ is sufficiently large, the complexities of
> > our expansion and compression modules are $O(d n K r)$ and $O(p n r)$,
> > respectively. However, when $n$ is relatively small, the cost of the
> > Cholesky decomposition, $O(\frac{1}{3} r^3)$, becomes non-negligible. This
> > is reflected in Figure 2a, where our method shows slightly higher complexity
> > than the baselines for small $n$.
> >
> > ## Writing
> > Thank you for your suggestions. We have made all possible corrections to
> > address the details you mentioned.
> >
> > Regarding the application domain, indeed, our method has the potential to be applied in other areas, such as natural language processing. This is an avenue we plan to explore in future work.
> >
> > ## Experiments
> >
> > **When and why would I use this method?**
> >
> > Indeed, our experiments were primarily designed to validate the theoretical
> > advantages of the method, rather than to immediately highlight specific
> > application scenarios.
> >
> > From the perspective of theoretical benefits and interpretability, our method
> > is grounded in a clear geometric structure, enabling layer-wise feature
> > expansion and compression in a principled way. This explicit geometric
> > formulation also opens the possibility of building logic-based operations on
> > top of the learned structures. We believe that such a principled and
> > interpretable design could facilitate the development of more trustworthy
> > and reliable artificial intelligence in the future.
> >
> > **Visualization on more layers.**
> > In Appendix C.6, we provide additional layer-wise visualizations, showing
> > that different heads in each layer indeed focus on different parts of the image.
> > It is worth noting that, due to our layer-wise compression mechanism,
> > attention maps in deeper layers may appear somewhat fragmented, as some
> > irrelevant features have already been discarded at this stage.

---

### Official Review · Reviewer_XL3D · 2025-11-01

**Soundness:** 2
**Presentation:** 3
**Contribution:** 2
**Rating:** 2
**Confidence:** 3

**Summary:**

This paper introduces ECAttention, an attention mechanism derived using the principle of Maximal Coding Rate Reduction. It models feature expansion and compression as operations that adaptively change the dimensionality of the feature subspace, where feature expansion is achieved by projecting features onto the null space of the overall data,  while compression is achieved by projecting features toward their respective class-specific column spaces.
To make this geometric operation computationally feasible, the paper uses a randomization method combined with Cholesky decomposition.

**Strengths:**

- Novelty of proposed attention like layer: The proposed ECAttention layer appears to be novel, fixing the flawed approximation in prior art.
- Promising Interpretability: Qualitative results provide evidence in favor of the intepretability of the proposed method.
- Potential scalability: The randomized approximation makes the proposed idea potentially scalable.

**Weaknesses:**

- Empirical Validation: As per table-1 ECA seems to trail TOST on the larger scale ImageNet/ImageNet Real datasets. Given the claimed superiority of the proposed method, I find this result quite surprising. While an explanation is provided in terms of the value of the parameter ‘r’, I find it unconvincing. It is on the authors to provide the relevant evidence for their claims. What is preventing the authors from exploring larger ‘r’ for larger models?
- Unclear Advantage over alternatives: While figure 2 aims to provide evidence in favor of ECA in terms of inference time and memory usage, especially in comparison to ToST, I find the evidence unconvincing. It is unclear from the presented results, if the proposed ECA method is preferable over ToST, and if yes, then in what scenarios/conditions.
- Missing comparison to non-whitebox approaches: Where does this work stand in comparison to data-driven non-whitebox approaches? It is not clear from the text how this paper pushes the frontier on the whitebox models, bringing them closer to SOTA (non-whitebox) method on the studied benchmarks.

Overall, I find the method rather ad-hoc, as it is based on a principle (MCR2) rather than being data driven. This by itself is not a major obstacle to acceptance, however, coupled with unconvincing empirical validation establishing superiority over alternatives (e.g. ToST, or even Imagenet classification in general), it is unclear why the proposed method would be of use to the community as a ready to use method or as a promising research direction.

**Questions:**

Please comment on the questions raised in the weaknesses section. Happy to be convinced otherwise.

---

> ### Author Response · Authors · 2025-11-21
>
> Thanks for your careful reading and valuable comments.
> We focus here on the issues raised regarding our experiments and provide detailed responses and corresponding revisions.
>
> **The superiority of our method over ToST and the empirical validation on hyperparameter r.**
> Compared with ToST, the superiority of our method lies in that the compression modules at each layer can capture correlation among dimensions for feature updates, achieving better interpretability (as evidenced by Figure6) and higher classification accuracy (as evidenced by Figure9), while incurring only a marginal increase in computational complexity compared to ToST.
>
> We find that **the strong performance of ToST reported in Table1 is largely attributable to its non-white-box MLP module**. This is also reflected in Table 8 of the ToST paper’s appendix[1]. We provide additional explanation in Section 4.2, and in Section 4.4, we include a fairer comparison with ToST.
> Specifically, the MLP module accounts for nearly two-thirds of ToST’s parameters. In Figure9, we show that removing the MLP module reduces ToST-S (22M) to 8M parameters (denoted as ToST-S-w/o MLP).
> Our proposed model, ECA-S (6M) with $r=10$ (accuracy on CIFAR-10 is 81.61\%), already surpasses the performance of ToST-S-w/o MLP (81.16\%). Moreover, increasing $r$ further improves accuracy, as evidenced by ECA-S with $r=40$, which achieves 82.14\% accuracy.
> As shown in Figure8b, by applying the softplus trick[2] to ensure positive weights for the compression and expansion modules, the trend of higher accuracy with larger $r$ becomes more stable.
>
> Due to time and computational resource limitations, we conducted the experiments with different  $r$ values on CIFAR-10 and CIFAR-100 (Appendix C.2). Performing these experiments on large-scale datasets such as ImageNet is infeasible with our current resources within a short timeframe. Nevertheless, we believe that the existing experimental results are sufficient to demonstrate that our method further advances this line of research and makes a meaningful contribution.
>
> **Compare to non-white-box approach**
> In our earlier version, we did not include this comparison because, as reported in the CRATE paper, ViT has a similar quadratic complexity to CRATE and achieves comparable performance. In contrast, our method significantly outperforms CRATE. Therefore, we did not provide a direct comparison with ViT in the previous version.
>
> In the revised version, we include the  results of the non-white-box method DeiT (a data-efficient Vision Transformer variant widely used as a baseline) in Figure 5. The figure shows that ToST still outperforms DeiT. As mentioned earlier, the superior performance of ToST is mainly attributed to its non-white-box MLP module.
> In contrast,
> our method advances the white-box line of research and makes a meaningful contribution, achieving performance comparable to the non-white-box method DeiT.
>
> **References :**
>
> [1] Wu, Ziyang, et al. "Token Statistics Transformer: Linear-Time Attention via Variational Rate Reduction." The Thirteenth International Conference on Learning Representations.
>
> [2] Dugas, Charles, et al. "Incorporating second-order functional knowledge for better option pricing." Advances in neural information processing systems 13 (2000).

---

> ### Comment · Reviewer_XL3D · 2025-11-28
> **Response to Author Rebuttal**
>
> I have read the author rebuttal, other reviews and corresponding author rebuttals. The general sense I get is that the paper remains unconvincing in terms of it's empirical validation and fails to establish clear advantage over alternatives. These concerns are common with most of the other reviews. While authors have provided responses to these concerns, I find the responses unconvincing and feel that more experimental works is needed to demonstrate the advantages, in addition to a significant rewrite to better motivate and explain the work.
>
> Following are some of the reasons for why I remain convinced
> 1. Interpretability: It is not clear what authors mean by interpretability. Looking at figure 6, it is not clear how ECA-B is more interpretable. ToST-S set of columns seem to also indicate object shapes, those in negative space. Overall, this figure fails to convince me that ECA is more interpretable than ToST.
> 2. Figure 9 is used to claim higher accuracy. However, these numbers are quite close (ECA-6M vs. ToST-8M) and without errors bars such differences on Cifar-10 are essentially meaningless. While the clipping of the bars on the left to 78 tends to amplify the difference, the absolute numbers remain too close to mean anything. Similarly, ECA-20M vs. ToST-S don't see to be different in accuracy as well. Overall, this figure fails to establish any advantage of EVA over ToST.
> 3. More time needed for experiments. This is a key point and I don't expect authors to run more experiments. All the major experimentation is supposed to be done in the initial version itself and rebuttal is mainly for additional clarifications and analysis of existing experiments. Overall, I feel that indeed more experimentation is needed and my recommendation would be to take that time and address the raised concerns for the next submission.
> 4. I appreciate the effort for providing DeIT comparison. However, this experimentation feels rushed (understandably). It is not clear why DeIT is a good comparison point for a non-whitebox method. Figure 5 seems to indicate ToST is already better than DeIT. My understanding is that whitebox methods are still behind whitebox methods. Overall, I find this experiment unconvincing.
>
> After taking everything into consideration, I remain convinced and don't think the paper is ready for acceptance.

---

> > ### Author Response · Authors · 2025-11-29
> >
> > Thanks for your reply.
> > We apologize if our previous rebuttals caused any misunderstanding.
> > We would like to clarify the point once again here.
> >
> >
> > **(1)Interpretability**. When we claim that our ECA model provides better interpretability, we mean that—compared with ToST—**our ECA model is more consistent with the principled theory of compressing features of different groups into different subspaces.
> > We note that this theoretical alignment was highlighted by the reviewer as a strength in the initial review.**
> > As shown in Figure 6, this is reflected by the fact that different heads in our model focus on different regions of the image, thereby capturing fine-grained structural information(e.g., separating a building's floors, or a tree from a door). In contrast, ToST only coarsely distinguishes between foreground and background.
> >
> > **(2)Claimed higher accuracy in Figure 9.**
> > We agree that the absolute accuracy differences in Figure 9 are small. However, the purpose of this figure is not to claim a large accuracy gain. Instead, Figure 9 is intended to illustrate two points:
> >
> > 1) Under a smaller parameter budget, ECA achieves comparable (and slightly higher) accuracy relative to ToST without the MLP module, while simultaneously exhibiting **significantly stronger and more principled subspace–separating interpretability**, as discussed in our earlier point. Therefore, this figure is intended as supportive evidence rather than a central contribution.
> >
> > 2) For the comparison between ECA-20M and ToST-S, the goal is also not to argue that ECA-20M substantially outperforms ToST-S in accuracy. Rather, ToST-S includes an MLP component that accounts for nearly **two-thirds of its parameters** and is not a white-box component. The intention of this comparison is to show that **even when ToST uses a large non–white-box MLP block, ECA still achieves comparable accuracy, while maintaining a fully interpretable architecture.**
> >
> > We will further consolidate our results and include error bars in the updated version to provide a more complete picture of performance variability.
> >
> > **(3) More time needed for experiments?**  We fully agree that the rebuttal period is not intended for running large new experiments. We would like to clarify that the major experiments supporting our core contributions—namely, the theoretical formulation of ECA, its principled **group-wise subspace–compressing behavior**, and comparisons with both ToST and CRATE—were already included in the initial submission. These experiments directly demonstrate the advantages and novelty of our interpretable framework.
> >
> > In particular, as shown in Appendix A.1 and Figure 6, ToST ignores the cross-dimensional structure of features, and its heads do not exhibit the intended subspace-compression behavior. In contrast, each head in ECA reliably implements this principled mechanism while maintaining comparable accuracy. Moreover, compared with CRATE, ECA enjoys clear advantages in terms of theoretical alignment, computational complexity, and performance.
> >
> > The additional results provided during rebuttal were included primarily to clarify why ToST appears stronger in Table 1 and Figure 5—namely, because a substantial portion of its performance is attributable to its non-white-box MLP module. This further highlights that our fully white-box method meaningfully advances this line of research and provides a valuable contribution to the field.
> >
> > **(4) DeiT comparison.** Our intention in including a comparison with DeiT is not to claim that ECA outperforms DeiT. Rather, DeiT serves as a standard and widely used baseline for Transformer-based image classification, and is routinely included in prior works such as ToST and CRATE. The comparison is therefore intended to provide context, situating white-box models in relation to a classical black-box Transformer.
> >
> > Regarding the reviewer’s point that Figure 5 shows ToST outperforming DeiT, this observation is fully consistent with our analysis: ToST’s superior accuracy largely stems from its non–white-box MLP module, which accounts for roughly two-thirds of its parameters. This reinforces our claim that ToST’s advantage does not come from its interpretable components.
> > It justifies using DeiT as a conventional reference baseline, allowing us to situate ECA and ToST (including the non-white-box MLP module) relative to a widely adopted black-box Transformer.
> >
> > Hence, the DeiT comparison is not rushed—it simply offers a conventional reference point, while our main contributions lie in the theoretical formulation of ECA and its principled group-wise subspace–compressing behavior.
> >
> > In summary, we believe that the current submission provides strong and sufficient evidence to support the core contributions of our fully white-box interpretable network, and we hope that this will be recognized in the evaluation.

---

> > > ### Author Response · Authors · 2025-11-30
> > >
> > > The updated versions of Figure 9 and Figure 12 (now including error bars) have been uploaded. The overall comparison trends remain consistent with those reported in the earlier version.

---

### Official Review · Reviewer_26TB · 2025-11-07

**Soundness:** 2
**Presentation:** 2
**Contribution:** 2
**Rating:** 4
**Confidence:** 3

**Summary:**

The authors propose a transformer architecture/parameterisation based on maximising the maximal coding rate reduction objective (MCR$^2$) of Chan et al. (2022). The objective posits that the transformer block encodes $K$ distinct clusters/downstream tasks, where $K$ is a user-specified hyperparameter (e.g., $K = 6$ in some of the experiments). Maximising the objective, defined as the mutual information $I[Z; \pi]$ between the representations $Z$ and their soft assignment $\pi$ to one of the $K$ classes, encourages the representations to be as informative as possible for each task.

The authors specifically assume that both $Z$ and $Z \mid \pi$ are Gaussian distributed, thus $I[Z; \pi]$ is available in closed form. In particular, the authors consider the decomposition $I[Z; \pi] = h[Z] - h[Z \mid \pi]$, which turns into a difference in the log-determinants of $Z$ and $(Z \mid \pi)$'s covariance matrices. Unfortunately, working with full covariance matrices would be computationally too expensive. Motivated by this challenge, the authors propose parameterising these covariance matrices in a way that balances representational capability and computational efficiency.

The authors perform qualitative and quantitative experiments on toy datasets. Concretely, they demonstrate that 1) their transformer architecture learns to extract more diverse and (hopefully) useful features than comparable methods and 2) for smaller model sizes, their method achieves higher accuracy than comparable methods.

## References

Kwan Ho Ryan Chan, Yaodong Yu, Chong You, Haozhi Qi, John Wright, and Yi Ma. Redunet: A white-box deep network from the principle of maximizing rate reduction.

**Strengths:**

The authors' approach is reasonably well-motivated and demonstrates reasonable performance. I particularly appreciated the quantitative comparisons in Figure 5, which show that their transformer architecture extracts more diverse and meaningful representations.

**Weaknesses:**

The paper's two main weaknesses are the lack of "exact" baselines in the experiments and the writing.

## The baseline

The authors only compare their method against other approximate methods. Concretely, they lack comparisons to 1) "standard" multi-head self-attention (MHA) transformers and 2) to ReduNet networks that do not introduce any approximations.
As such, it is difficult to ascertain
 1. how much benefit does the authors' method bring compared to just using "standard" MHA and
 2. how severe is the approximation that they make?

Including these baselines would significantly strengthen the paper.

## The writing

Generally, there are two issues with the writing.
First, the authors assume the reader is far too familiar with ReduNet and related approaches, which is reflected in the writing in various ways:
- The Introduction section is already too much of a literature review; I was lost by the end of the first paragraph, as it essentially consists of an enumeration of competing methods.
- Abbreviations such as ToSt, CRATE, and TSSA are undefined.
- There is no discussion of the MCR$^2$ objective; where it comes from, what it means, etc. I could only understand the motivation after skimming the original ReduNet paper.
- Across the main text, there are several places where the authors explain how their method works better than competing methods, but those competing methods are never introduced. As such, these comparisons are lost on readers like me, who are unfamiliar with them.
- At the same time, several interesting and relevant details are omitted or relegated to the appendix. For example, why is the direct column-space projection more numerically unstable than subtracting the null null space projection? What purpose do the $U_k$ terms serve?

Second, despite the authors' claim that their method is principled, I did not find that their exposition bore this out. Generally, the authors should spend more time motivating and explaining the objective function and less time comparing their method to competing ones if they do not intend to describe them in sufficient detail.

Miscellaneous:
 - "inter-dimensional correlations" -- no need for the adjective
 - "first tokenized into vectors X= [x1,...,xn] ∈RD×n (i.e., tokens)" -- repetition
 - Why does "multi-head self-attention operator" abbreviate to "MSSA"?
 - Eq 3: bottom row of column vector on right-hand side: should be $U_K Z^\top$ instead of $U_1 Z^\top$.
 - Figure labels are too small and are unreadable without zooming in; please increase font size.

**Questions:**

n/a

---

> ### Author Response · Authors · 2025-11-21
>
> Thank you for the valuable feedback on the baselines and writing. We address your concerns point-by-point below.
>
> # Comments on Baseline
>
> **Compare with "standard"multi-head self-attention transformers.**
> In our earlier version, we did not include this comparison because, as reported in the CRATE paper, ViT has a similar quadratic complexity to CRATE and achieves comparable performance. In contrast, our method significantly outperforms CRATE. Therefore, we did not provide a direct comparison with ViT in the previous version.
>
> In the revised version, we incorporate the results of DeiT (a data-efficient Vision Transformer variant) in Section 4.2. Our proposed ECA model advances the performance of current white-box methods and achieves comparable performance to the non-white-box method DeiT.
> While Figure 5 shows that ToST outperforms DeiT on the large-scale ImageNet dataset, our further investigation reveals that ToST's superior performance primarily stems from its MLP module, which accounts for approximately 2/3 of the total model parameters. This is also reflected in Table 8 of the ToST original paper’s appendix [1]. We provide a fairer comparison in Section 4.4 to demonstrate the superiority of our method.
>
> **Comparison with ReduNet: severity of approximation**
> ReduNet cannot be directly applied to large datasets---even datasets such as CIFAR-10.
> The toy-data results shown in Figure (3) already demonstrate that our proposed method is capable of achieving layer-wise compression and expansion.
>
> ReduNet only includes a forward layer-construction procedure without any backpropagation-based training. Moreover, each layer must be constructed using all training samples and requires cubic computational complexity. As a result, ReduNet cannot be directly applied to large datasets. The experiments reported in the original ReduNet paper on CIFAR-10 and other image datasets were actually obtained using a ResNet backbone with $\text{MCR}^2$ objective.
>
> In addition, our optimization objective differs from the original ReduNet formulation. Our network architecture is derived from the modified compression objective that incorporates the $K$ subspaces $U_[K]$, as described in Eq. (2). Therefore, following the common practice in this line of research, such as CRATE and ToST, we focus our comparison on the methods most relevant to our approach and exclude ReduNet results.
>
> # Comments on Writing
> **Assume excessive prior knowledge.**
> We sincerely thank the reviewer for this valuable feedback and apologize that the writing assumes too much prior knowledge, especially in the introduction.
> However, we found it necessary to briefly summarize the prior white-box models (e.g., ReduNet, CRATE, ToST) in order to properly motivate our idea.
> For clarity, we address the issues you raised and have made the corresponding revisions in the manuscript:
>
> (1) **Related works and abbreviations.** The progression of this research direction is as follows: ReduNet $\rightarrow$ Coding-RATE transformer (CRATE, employing the multi-head subspace self-attention (MSSA)) $\rightarrow$ Token statistics transformer (ToST, employing the token statistics self-attention(TSSA)).
>
> (2) **Discussion of $\text{MCR}^2$.**
> We provide the formal definition of $\text{MCR}^2$ in Section 2 (Eq. 1) and explain its underlying intuition.
> Briefly, the objective consists of two components:
> 1) The first term $R(Z)$ of Eq. 1 encourages the span of all feature vectors to be sufficiently large (i.e., high-dimensional).
> This ensures that the learned representations contain rich information across multiple dimensions, which also facilitates the separation of different class subspaces in a high-dimensional space.
> 1) The second term $R_c(Z,\Pi)$ of Eq. 1 encourages the features of each individual class subspace to be tightly compressed into a low-dimensional structure. This is precisely the intuition behind the MCR objective.
>
> (3) **Introduction of competing methods.**
> Perhaps the reviewer did not notice that Section 2 of our paper is precisely devoted to introducing the competing methods.
> In particular, right below Eq. 3, we describe the weaknesses of the MSSA module in CRATE and the TSSA module in ToST.
> The structure of the MSSA module is derived from a second-order approximation of the gradient of the compression objective in Eq. 2. However, this approximation discards the first-order term, which causes MSSA to violate the intended compression goal.
> Additionally, the TSSA module is derived from the gradient of an upper-bound surrogate of the compression objective in Eq. 2. It depends only on the diagonal elements of the feature correlation matrix and ignores correlations among dimensions, which limits its ability to capture the fine-grained intrinsic structure of the data.
> We also provide the corresponding theoretical analysis in Appendix A.1.

---

> ### Author Response · Authors · 2025-11-21
>
> (4) **Why is subtracting the null space projection stable?**
> The reason is that compression via subtracting the null-space projection yields a skip-connection–like update, i.e., $z_{\ell+1} =z_{\ell} - \text{(null-space projection)}$, which is more favorable for multi-epoch and layer-wise optimization.
> In contrast, directly projecting onto the column space results in a structure like $z_{\ell+1} =  \text{(column-space projection)}$.
> This structure blocks the input information of each layer from flowing to subsequent layers, which undermines both layer-wise optimization and multi-epoch training.
>
> We provide additional clarification on this point in Section 4.1.
> Figure 4 presents a comparison between compression achieved by direct projection onto the column space and compression performed by subtracting the projection onto the null space.
> Compression by subtracting null space projection allows effective optimization.
>
>
> (5) **What purpose do the $U_k$ terms serve?**
> We can view  $U_k$ as the $k$-th low-dimensional subspace.
> In summary, the compression objective aims to project features from different classes into separate subspaces.
> The difference lies in how the subspaces are defined: in ReduNet, each class subspace is formed by the features of that class itself, whereas in CRATE and our work, the $K$ subspaces are represented by $K$ matrices $U_{[K]}$.
> For example, in the case of images, each layer can compress features of different types into $K$ heads/subspaces.
> We have revised the description above Eq.(2) in Section 2 to clarify the role of $U_k$.
>
> (6) **The principled method?**
> In this paper, we use the term *principled* to indicate that the
> network architecture and its layer-wise update rules can be
> derived from a well-defined optimization objective. Specifically, our
> method is grounded in the expansion objective $R(Z)$ in Eq. (1) and the compression objective in Eq. (2). Under this definition,
> a principled feature update should follow the gradient (or a faithful
> approximation of the gradient) of these objectives, ensuring that the
> update direction is theoretically justified rather than heuristic.
> In our paper, we construct the expansion and compression modules at each
> layer based on the geometric interpretation of the gradient of the
> corresponding objectives. This design enables layer-wise feature expansion
> or compression, as illustrated in Figure (3).
>
> All other miscellaneous issues you mentioned have been addressed in the main text. Additionally, we have increased the font size in the figures for better readability.
> Thanks for all your suggestions.
>
>
> **References :**
>
> [1] Wu, Ziyang, et al. "Token Statistics Transformer: Linear-Time Attention via Variational Rate Reduction." The Thirteenth International Conference on Learning Representations.

---

### Author Response · Authors · 2025-11-21

We thank all reviewers for their valuable feedback and for recognizing the contributions of our work. The main concerns shared by most reviewers are the comparison with non–white-box methods and the reason why our method performs slightly worse than ToST in Table 1. To address these questions, we provide the following clarifications and additional experiments:

In our earlier version, we did not include this comparison because, as reported in the CRATE paper, ViT has a similar quadratic complexity to CRATE and achieves comparable performance. In contrast, our method significantly outperforms CRATE. Therefore, we did not provide a direct comparison with ViT in the previous version.

In Figure 5 and Appendix B.3 of the revised version, we include results on DeiT (a  ViT variant that is widely used as a baseline) to demonstrate that our proposed method improves upon existing white-box models and achieves performance comparable to DeiT.

The superior performance of ToST primarily comes from its non–white-box MLP module, which accounts for nearly two-thirds of its parameters.
This is also reflected in Table 8 of the ToST original paper’s appendix [1].
In Section 4.4, we provide a fairer comparison by controlling this factor, showing the advantage of our approach.

For the other concerns raised by the reviewers, we address each of them in the corresponding responses.
All figure/table numbers cited below are based on the revised paper.


References :

[1] Wu, Ziyang, et al. "Token Statistics Transformer: Linear-Time Attention via Variational Rate Reduction." The Thirteenth International Conference on Learning Representations.

---

### Author Response · Authors · 2025-11-30

Dear Area Chair,

Thank you for coordinating the review of our paper.
We write to summarize the discussion with reviewers and highlight that the primary concerns have been resolved.

### Main Contribution:

Previous white-box methods (CRATE, ToST) fall short of principled and effective compression: CRATE incurs quadratic complexity and deviates from its compression objective by discarding the first-order approximation, while ToST neglects correlations among dimensions during feature updates.
To address these limitations,
this paper proposes a **fully principled, nearly linear-time** attention mechanism inspired by the geometric insight of the maximal coding rate reduction ($\text{MCR}^2$) objective function.
Specifically, we leverage a randomness approach to effectively capture the low-dimensional column space of features, thereby compressing group-wise features into their  corresponding low-dimensional subspaces.

### Reviewers' Agreement on Core Contribution:

All reviewers consistently acknowledge the paper's core contribution: **the proposed ECAttention performs effective and principled subspace compression, while offering significantly improved interpretability compared to ToST**. Reviewers agree that, unlike ToST---which ignores correlations among dimensions---ECAttention provides a theoretically grounded mechanism that preserves meaningful structural relationships during compression.

This consensus is further supported by the qualitative evidence presented in **Figure 6** of the revised manuscript (formerly Figure 5 in the initial submission). The visualizations demonstrate that ECAttention produces **fine-grained and semantically coherent decompositions** of images.

In particular, reviewer **vfW8** explicitly emphasized the strength of our interpretability results:

> *"The membership visualizations for ECA-B are **qualitatively excellent**, demonstrating a fine-grained, semantically meaningful segmentation of objects (e.g., separating a building's floors, or a tree from a door). This provides strong evidence for the authors' claim that ToST's simplification (ignoring inter-dimensional correlations) misses important structural information, and that ECAttention's approach successfully captures it."*

This direct reviewer endorsement strongly supports our main claim that **ECAttention maintains principled compression while capturing richer structural information than ToST**, validating both the motivation and the impact of our method.

---

> ### Author Response · Authors · 2025-11-30
>
> ### Main concerns:
> There are two concerns shared across all reviewers in the initial  submission:
>
> **Concern 1: Why does ToST achieve better accuracy?**
>
> **Response:**
> The strong performance of ToST reported in Table 1 and Figure 5 in revised version is **largely attributable to its non-white-box MLP module.** This is also reflected in Table 8 of the ToST paper's appendix [1].
> We addressed this concern by including a fairer comparison experiments with ToST in Section 4.4.
> Specifically, the MLP module accounts for nearly two-thirds of ToST's parameters. In Figure 9, we show that removing the MLP module reduces ToST-S (22M) to 8M parameters (denoted as ToST-S-w/o MLP).
> Our proposed model, ECA-S (6M) with $r=10$ (accuracy on CIFAR-10 is 81.61%), already surpasses the performance of ToST-S-w/o MLP (8M, 81.16%).
> Moreover, increasing hyperparameter $r$ further improves accuracy, as evidenced by ECA-S with $r=40$, which achieves 82.14% accuracy.
>
> **The Reviewer vfW8 acknowledged our response**, stating:
> > *"The authors have effectively addressed my primary concerns regarding the technical stability and comparative fairness of the proposed method."*
>
> **Notably, vfW8 had indicated willingness to raise the score in the initial review.**
>
> However, **Reviewer XL3D** further questioned the significance of the accuracy differences shown in Figure 9.
>
> **Response:**
> We agree that the absolute accuracy differences in Figure 9 are small. However, the purpose of this figure is not to claim a large accuracy gain. Instead, Figure 9 is intended to illustrate two points:
>
> 1) Despite having fewer parameters, ECA achieves comparable or marginally better accuracy than ToST when the MLP module is removed, while simultaneously exhibiting **significantly stronger and more principled subspace–separating interpretability, which is our core contribution that agreed by all reviewers**. Therefore, the Figure 9 is intended as supportive evidence rather than a central contribution.
>
> 2) For the comparison between ECA-20M and ToST-S, the goal is also not to argue that ECA-20M substantially outperforms ToST-S in accuracy.
> Rather, ToST-S includes an MLP component that accounts for nearly two-thirds of its parameters and is not a white-box component. The intention of this comparison is to show that **even when ToST uses a large non–white-box MLP block, ECA still achieves comparable accuracy, while maintaining a fully interpretable architecture.**
>
> We will further consolidate our results and include error bars in Figure 9 in the updated version to provide a more complete picture of performance variability.
>
> **Concern 2: Compared with non-white-box method.**
>
> **Response:** In our earlier version, we did not include this comparison because, as reported in the CRATE paper, ViT has a similar quadratic complexity to CRATE and achieves comparable performance. In contrast, our method significantly outperforms CRATE. Therefore, we did not provide a direct comparison with ViT in the previous version.
>
> In Figure 5 and Appendix B.3 of the revised version, we include results on DeiT (a ViT variant that is widely used as a baseline) to demonstrate the relative differences between existing white-box methods and DeiT.
>
> However, the **Reviewer XL3D** further questioned the reason of using DeiT as baseline.
>
> **Response:**
> Our intention in including a comparison with DeiT is not to claim that ECA outperforms DeiT. Rather, DeiT serves as a standard and widely used baseline for Transformer-based image classification, and is routinely included in prior works such as ToST and CRATE. The comparison is therefore intended to provide context, situating white-box models in relation to a classical black-box Transformer.
>
> Regarding the reviewer's point that Figure 5 shows ToST outperforming DeiT, this observation is fully consistent with our analysis: **ToST's superior accuracy largely stems from its non–white-box MLP module** , which accounts for roughly two-thirds of its parameters. This reinforces our claim that ToST's advantage does not come from its interpretable components. It justifies using DeiT as a conventional reference baseline, allowing us to situate ECA and ToST (including the non-white-box MLP module) relative to a widely adopted black-box Transformer, **while our main contributions lie in the theoretical formulation of ECA and its principled group-wise subspace–compressing behavior**.
>
> In summary,  all reviewers have acknowledged our core contributions.
> We believe we have adequately addressed the two main concerns shared by the reviewers.
> The current submission provides strong and sufficient evidence to support the core contributions of our fully white-box interpretable network. We respectfully hope that the Area Chair will consider these points in the final evaluation.

---

> > ### Author Response · Authors · 2025-11-30
> >
> > ### References:
> >
> > [1] Wu, Ziyang, et al. "Token Statistics Transformer: Linear-Time Attention via Variational Rate Reduction." The Thirteenth International Conference on Learning Representations.

---

> > > ### Author Response · Authors · 2025-11-30
> > >
> > > The updated versions of Figure 9 and Figure 12 (now including error bars) have been uploaded. The overall comparison trends remain consistent with those reported in the earlier version.

---

### Meta-Review · Area_Chair_VLYQ · 2026-01-04

**Summary:**

Reviewer 26TB: The authors' approach is reasonably well-motivated and demonstrates reasonable performance with quantitative comparisons showing that their transformer architecture extracts more diverse and meaningful representations. However, the reviewer still has some concerns on the weaknesses about the lack of  baselines in the experiments and the writing isues.

Reviewer XL3D:  The proposed ECAttention layer appears to be novel, fixing the flawed approximation in prior art. Qualitative results provide evidence in favor of the intepretability of the proposed method. The randomized approximation makes the proposed idea potentially scalable. However, the reviewer still has some concerns on the weaknesses about the specific empirical validation, unclear advantage over alternatives, missing comparison to non-whitebox approaches.

Reviewer 6NA9: Several motivations of this work are clear. The results are competitive, yet the training is fast. The reviewer still has some concerns on the weaknesses about the confusing notations, unsupported claims, unfit for ICLR, not compelling experiments.

Reviewer vfW8: The paper's strongest point is Figure 5. The membership visualizations for ECA-B are qualitatively excellent, demonstrating a fine-grained, semantically meaningful segmentation of objects. The paper plots inference time and peak memory vs. tokens and provides an appendix complexity breakdown. However, the reviewer still has some concerns on the weaknesses about the unclear surpass over the strongest baselines at scale, no evaluation on long-sequence tasks.

**Reviewer Concerns:**

After carefully evaluating the rebuttals, I think the reviews from the Reviewer  vfW8 were partially addressed from the response.
For the remaining reviewer concerns, they are all not fully addressed.

**Reviewer Scores:**

For the Reviewer Reviewer 26TB, 6NA9 and XL3D, I think the reviewer may keep the rating unchanged based on the response.

For the  Reviewer vfW8, I think the reviewer may increase the rating or keep the rating unchanged based on the response.

---

### Decision · Program_Chairs · 2026-01-26

Reject